# Cytoplasmic Interactions between the Glucocorticoid Receptor and HDAC2 Regulate Osteocalcin Expression in VPA-Treated MSCs

**DOI:** 10.3390/cells8030217

**Published:** 2019-03-05

**Authors:** Marcella La Noce, Luigi Mele, Luigi Laino, Giovanni Iolascon, Gorizio Pieretti, Gianpaolo Papaccio, Vincenzo Desiderio, Virginia Tirino, Francesca Paino

**Affiliations:** 1Department of Experimental Medicine, University of Campania “Luigi Vanvitelli”, 80138 Naples, Italy; marcella.lanoce@unicampania.it (M.L.N.); luigi.mele@unicampania.it (L.M.); vincenzo.desiderio@unicampania.it (V.D.); virginia.tirino@unicampania.it (V.T.); 2Multidisciplinary Department of Medical-Surgical and Odontostomatological Specialties, University of Campania, “Luigi Vanvitelli”, 80121 Naples, Italy; luigi.laino@unicampania.it (L.L.); dott.goriziopieretti@gmail.com (G.P.); 3Department of Medical and Surgical Specialties and Dentistry, University of Campania “Luigi Vanvitelli”, 80121 Naples, Italy; giovanni.iolascon@unicampania.it; 4Department of Biomedical, Surgical and Dental Sciences, University of Milan, 20133 Milan, Italy

**Keywords:** epigenetic drug, glucocorticoid receptor, HDAC2, human mesenchymal stem cells, osteogenic differentiation

## Abstract

Epigenetic regulation has been considered an important mechanism for influencing stem cell differentiation. In particular, histone deacetylases (HDACs) have been shown to play a role in the osteoblast differentiation of mesenchymal stem cells (MSCs). In this study, the effect of the HDAC inhibitor, valproic acid (VPA), on bone formation in vivo by MSCs was determined. Surprisingly, VPA treatment, unlike other HDAC inhibitors, produced a well-organized lamellar bone tissue when MSCs–collagen sponge constructs were implanted subcutaneously into nonobese diabetic/severe combined immunodeficiency (NOD/SCID) mice, although a decrease of osteocalcin (OC) expression was observed. Consequently, we decided to investigate the molecular mechanisms by which VPA exerts such effects on MSCs. We identified the glucocorticoid receptor (GR) as being responsible for that downregulation, and suggested a correlation between GR and HDAC2 inhibition after VPA treatment, as evidenced by HDAC2 knockdown. Furthermore, using co-immunoprecipitation analysis, we showed for the first time in the cytoplasm, binding between GR and HDAC2. Additionally, chromatin immunoprecipitation (ChIP) assays confirmed the role of GR in OC downregulation, showing recruitment of GR to the nGRE element in the *OC* promoter. In conclusion, our results highlight the existence of a cross-talk between GR and HDAC2, providing a mechanistic explanation for the influence of the HDAC inhibitor (namely VPA) on osteogenic differentiation in MSCs. Our findings open new directions in targeted therapies, and offer new insights into the regulation of MSC fate determination.

## 1. Introduction

Bone is the major support tissue in the body, but it can be compromised by degenerative diseases or trauma. Therefore, research into bone regeneration remains of great interest, and it becomes a serious challenge within the field of regenerative medicine [1,2,3].

Currently, bone grafting is the “gold standard” method that is used to treat damaged or missing bone [4]; nevertheless, it does not achieve effective bone regeneration [5]. Additional concerns with this methodology include relevant cost and the high risks associated with a surgical procedure [6].

Over the past few years, the potential of using human mesenchymal stem cells (hMSCs) to regenerate different tissue types has been highlighted, due to the cells’ inherent capability of committing into different types of mature cells, such as osteoblasts or chondrocytes, among others [7,8]. The differentiation of hMSCs into bone-forming cells has also been reported. Dental pulp represents a promising source of mesenchymal stem cells (MSCs) for bone-replacement therapies [9]. Dental pulp stem cells (DPSCs) have received extensive attention in the field of bone tissue engineering, due to their distinct capability for differentiation into the osteogenic lineage [10,11,12], as well as into other cell types [13,14]. The osteoblastic potential of DPSCs has been widely demonstrated, both in vitro and in vivo, as well as after grafting in humans [15,16]. Bone lineage commitment is driven by the expression of the transcription factor RUNX2 [17], which further promotes the expression of the early markers Alkaline phosphatase (ALP) [18], Osterix (OSX), intermediate and late markers, including Collagen type I, Osteopontin (OPN), Bone sialoprotein (BSP), and Osteocalcin (OC). This sequential upregulation leads to osteoblast maturation and deposition of mineralized extracellular matrix [19].

The critical issue for the application of stem cells in tissue engineering is the control of cell differentiation. The regulation of mechanisms underlining stem cell state and/or differentiation is essential for the development of stem cell-based therapy. The events that govern stem cell differentiation are predominantly epigenetics. Indeed, specific patterns of DNA methylation and histone modifications play an important role in the induction of MSC differentiation toward specific lineages. Histone acetylation is the most well-studied histone modification, and it has been shown to be an important process of gene regulation. Histone acetylation contributes to the formation of a more relaxed, and thus transcriptionally active chromatin structure. In contrast, histone deacetylation is associated with a condensed chromatin form, leading to transcriptional repression. The level of cellular histone acetylation is regulated by the opposing activities of histone acetyltransferases (HAT) and histone deacetylases (HDACs), which reversibly catalyze the acetylation and deacetylation mechanisms, respectively [20,21]. HDACs have been shown to regulate several important biological processes, including cell proliferation, differentiation, and development, by forming complexes with various transcription factors and transcriptional coregulators [22]. The inhibition of HDAC activity typically leads to de-repression of transcription. HDAC inhibitors (HDACi) are natural or synthetic small molecules that can inhibit the activities of HDACs and promote efficient and temporally regulated control of gene expression. HDACi have shown very potent effects in stem cell differentiation pathways. They may promote either self-renewal [23,24] or differentiation [25,26] depending on the stem cell status, the dose employed and the cell type [27]. Therefore, application of epigenetic regulators, such as HDAC inhibitors, may be valuable for stem cell-based interventions. There is growing evidence that some HDACi also stimulate the osteogenic differentiation of MSCs [28,29,30,31,32].

In a previous study, we demonstrated in vitro that DPSCs pretreated with valproic acid, an HDAC inhibitor, significantly improve mineralized matrix formation, enhancing the expression of bone glycoproteins, such as *OPN* and *BSP*—both of which are involved in the formation of the mineralized matrix—but which negatively affect the expression of *OC*, a late-stage marker of differentiation. This downregulation was strongly correlated with the inhibition of HDAC2 [33]. The molecular mechanisms and key mediators of this effect are still unknown.

In this study, we have evaluated the effects of different HDAC inhibitors such as Valproic acid (VPA), Entinostat (MS-275), Trichostatin A (TSA) and Vorinostat (SAHA) on bone formation in vivo, and have demonstrated that only the VPA treatment produces well-organized lamellar bone tissue, although a decrease of osteocalcin (OC) expression occurred. Consequently, we decided to investigate the molecular mechanisms by which VPA acts in exerting those effects on MSCs. We identified, among the possible transcription factors involved in osteocalcin downregulation, the glucocorticoid receptor (GR), which is a ligand-inducible transcription factor belonging to the nuclear receptor superfamily.

In the absence of its ligand, GR resides in the cytoplasm as a multimeric chaperone complex with heat-shock protein 70 (hsp70), hsp90, p23, and immunophilins, among other factors. Upon ligand binding, GR dissociates from this complex, translocates into the nucleus, and positively regulates transcription by directly binding to specific glucocorticoid response elements (GREs) in the promoter region of its target genes, or as a monomeric protein that co-operates with other transcription factors to induce transcription [34,35,36]. Negative regulation also occurs when GR binds to negative glucocorticoid response elements (nGREs) having a consensus GRE sequence of 5′ GGTACAnnnTGTTCT 3′ from the transcription start site of the promoter [36]. In osteoblasts, the early bone-specific marker *RUNX2*, the late marker *OC* [37,38,39], and *BSP* [40] are among the direct targets of GR. It was found that GR inhibits *OC* through the nGREs on the distal region of the *OC* promoter [37,38]. Osteocalcin is a late marker of osteogenic differentiation. During bone development, there is little osteocalcin production, and it does not reach maximal levels until the late stages of mineralization. Osteocalcin binds to hydroxyapatite only in post-proliferative mature osteoblasts that are associated with mineralized osteoid [41,42].

In the present study, we demonstrate that VPA treatment on DPSCs is able to produce a well-organized bone tissue structure in vivo, although OC expression is decreased. Furthermore, we identified a correlation between GR and HDAC2 inhibition after VPA treatment that affects osteocalcin expression in DPSCs. Chromatin immunoprecipitation (ChIP) assays showed a recruitment of GR to the nGRE element in the *OC* promoter in DPSCs. In addition, we provide new evidence that HDAC2 is associated with GR in the cytoplasm.

## 2. Materials and Methods

### 2.1. Human Dental Pulp Extraction and Cell Culture

Human dental pulps were extracted from teeth of healthy adults (21–38 years of age, both male and female). Prior to the extraction, each subject (*n* = 40) was checked for systemic and oral infections or diseases. Only patients undergoing a third molar or supernumerary tooth extraction were interviewed and enlisted. All subjects signed the Ethical Committee (Second University Internal Ethical Committee) consent brochure before being enrolled. Every subject was pretreated for a week with professional dental hygiene. The dental crown was covered with 0.3% chlorhexidine gel (Forhans, New York, NY, USA) for 2 min prior to the extraction. Dental pulp was obtained with a dentinal excavator or a Gracey curette. The pulp was delicately removed and immersed for 1 hr at 37 °C in a digestive solution composed of 3 mg/mL type I collagenase and 4 mg/mL dispase in phosphate-buffered saline (PBS) containing 40 mg/mL gentamicin. Once digested, the solution was filtered through 70 μm Falcon strainers (Becton & Dickinson, Franklin Lakes, NJ, USA). Cells were cultured in basal growth medium consisting of Dulbecco’s modified Eagle’s medium (DMEM) with 100 U/mL penicillin, 100 mg/mL streptomycin, and 200 mM L-glutamine (all from GIBCO, Monza, Italy), supplemented with 10% fetal bovine serum (C-FBS; GIBCO, Monza, Italy). Cultures were maintained in a humidified atmosphere under 5% CO_2_ at 37 °C.

Human dental pulp stem cells (hDPSCs) were selected and characterized as previously described (La Noce et al, 2014). Briefly, flow cytometry analyses were performed on hDPSCs at the first passage of culture (approximately 1 × 10^6^ cells). Human DPSCs were sorted for CD34 and CD90 positive markers using a Fluorescence Activated Cell Sorting (FACS) Aria III BD (BD Biosciences, Milan, Italy). The purity of sorting was approximately 90%. For phenotypic characterization, cells were incubated with Fluorescein isothiocyanate (FITC)-conjugated anti-CD90, PerCP-Cy5.5-conjugated anti-CD105, APC-Cy7-conjugated anti-CD45 (all purchased from BD Pharmingen), and PE-conjugated anti-CD34 (Miltenyi Biotech) and FITC-conjugated anti-bone sialo-protein (BSP) (Biorbyt), anti-CFS-conjugated anti-osteopontin (OPN) (R&D Systems) for the evaluation of osteogenic differentiation. As negative controls, cells were stained with an isotype control antibody.

### 2.2. Chemicals and Reagents

For osteogenic differentiation, when cells at the third passage of culture reached 60–70% confluency, they were induced using osteoinduction medium, composed of DMEM supplemented with 10% FBS, 1% Pen-Strept, 50 μg/mL ʟ-ascorbic acid (Sigma, Gillingham, Dorset, UK), 10 mM glycerol phosphate disodium salt (β-glycerophosphate), and 10 nM dexamethasone (Sigma, Gillingham, Dorset, UK). Cells maintained in the basal culture medium served as the controls. The osteogenic medium was changed twice a week.

Valproic acid sodium salt, MS-275, TSA, SAHA (HDAC inhibitors) and RU-486 (Mifepristone, GR antagonist) were purchased from Sigma.

Cells were treated with 1.5 mM of VPA, 2.5 µM of MS275, 100 nM of TSA, and 1 µM of SAHA for 48 hours.

### 2.3. MTT Analysis

Cell viability was measured by the colorimetric 3-(4,5-dimethyl-2-thiazolyl)-2,5-diphenyltetrazolium bromide (MTT) assay. Cells were seeded in 96-well plates at a density of 10^4^ cells per well, then they were treated with different concentration of HDACi for 24 hr and 48 hr. After treatments, 100 μL of 1 mg/mL MTT (Sigma) in DMEM medium containing 10% FBS was added to the well for 4  hr at 37  °C. The medium was then replaced with 200 μL of Dimethyl sulfoxide (DMSO) and shaken for 15 min, then absorbance at 540 nm was measured using a microplate enzyme-linked immunosorbent assay (ELISA) reader with DMSO used as the blank.

### 2.4. RNA Extraction and qRT-PCR

Total RNA was extracted from hDPSCs using an AMBION kit (Life Technologies Italia, Monza, Italy) following the manufacturer’s instructions. RNA was treated with DNase (Promega, Milan, Italy) to exclude DNA contamination and stored at −80 °C. Complementary DNA (cDNA) synthesis was carried out from total RNA (1 μg) using GoScript Reverse Trancriptase (Promega, Milan, Italy). Samples were analyzed using real-time quantitative PCR (qRT-PCR). PCR reactions were performed using StepOne Thermocycler (Applied Biosystems, Monza, Italy) and the amplifications were done using the SYBR Green PCR Master Mix (Applied Biosystems, Monza, Italy). The thermal cycling conditions were: 50 °C for 2 min followed by an initial denaturation step at 95 °C for 2 min, 40 cycles at 95 °C for 30 s, 60 °C or 58 °C for 30 s and 72 °C for 60 s. Real-time PCR was performed using the primer sequences shown in Table 1. An additional step starting from 60 to 95 °C (0.05 °C·s^−1^) was performed to establish a melting curve. This was used to verify the specificity of the qRT-PCR reaction for each primer pair. For each measurement, a threshold cycle value (Ct) was determined. This was defined as the number of cycles necessary to reach a point at which the fluorescent signal is first recorded as being statistically significant above the background. Data were analyzed by using the 2^−ΔΔCt^ method to obtain the relative expression level, and each sample was normalized by using GAPDH RNA expression. The experiments were carried out in triplicate for each data point.

### 2.5. In Vivo Grafting

In order to achieve 3D tissue formation, treated and untreated cells were seeded on a Gingistat^®^ (GABA VEBAS) scaffold. This scaffold is a lyophilized collagen type I sponge that was cut under sterile conditions into 5 × 5 × 5 mm cubes. Briefly, scaffold cubes were placed in six-well plates, and a cell suspension of 1 × 10^6^ cells contained in 200 µL medium was pipetted onto the top of each cube. Cells were allowed to adhere under a humidified atmosphere at 37 °C and 5% CO_2_ for 4 hours. The seeded scaffolds were then placed in tubes containing osteogenic medium for 15 days. Osteogenic medium was changed twice a week.

Twenty-four female NOD/SCID mice, six weeks old, were purchased from Charles River (Charles River Laboratories International, Inc, Milan, Italy) and acclimatized for a week prior to experimentation. The seeded scaffolds (constructs) were implanted into subcutaneous dorsal pockets of the immunodeficient mice. In particular, four mice per group were transplanted with constructs of the cells treated with HDACi (VPA, MS275, TSA, SAHA) seeded on the scaffold; cells seeded on the scaffold but cultured in basal medium, and cells without HDACi treatment but induced in osteogenic medium were used as negative and positive controls, respectively. After 60 days, mice were sacrificed, and the constructs were removed, fixed in buffered formalin, and subsequently analyzed by immunohistochemistry.

### 2.6. Ethics Statement

All animal experiments were conducted in full compliance with the University of Campania “Luigi Vanvitelli”, and Italian Legislation for Animal Care (Permit Number: 440/2016-PR).

### 2.7. Cryostat Sectioning

Constructs removed from the subcutaneous excision were fixed in 4% paraformaldehyde (PFA) and cryoprotected overnight at 4 °C by immersion in a 30% (*w*/*v*) sucrose solution before being embedded in Tissue-Tek^®^ O.C.T. Compound (Tissue-Tek; Sakura Finetek, Torrance, CA, USA) and frozen. Sections 5 μm thick were cut with a cryostat at −20 °C, and then processed for immunostaining.

### 2.8. Immunohistochemistry/Immunofluorescence Analysis

Frozen sections were stained by Hematoxylin and Eosin to determine morphologic tissue organization.

Osteogenic differentiation was evaluated by Alizarin Red staining to visualize the calcium-rich deposits produced by the cells. The samples were washed in ddH_2_O to remove O.C.T. for 5 min, and then stained with Alizarin Red solution (2%, pH 4.2; Sigma Aldrich) for 20 min at room temperature. Stained samples were extensively washed with deionized water to remove any nonspecific precipitation. Micrographs were taken with a microscope Eclipse TE2000-S (Nikon, Firenze, Italy) and a Nikon camera (Nikon, Firenze, Italy).

For immunofluorescence staining, frozen sections were permeabilized with 0.1% Triton X-100 for 15 min. Each sample was incubated in PBS containing 5% bovine serum albumin (BSA) for 30 min. at room temperature as a blocking step. Then, after washing twice with PBS, samples were incubated with primary antibodies: rabbit polyclonal to osteopontin (1:1000, Abcam, Cambridge, UK), mouse monoclonal to osteocalcin (1:100, Santa Cruz, Heidelberg, Germany), mouse monoclonal to human nuclear antigen (1:100, Abcam, UK) overnight at 4 °C in the dark. This step was followed by incubation with the secondary antibody Tetramethylrhodamine (TRITC)-conjugates and FITC-conjugate (1:1000, Abcam). Nuclear counterstaining was performed with 4,6-diamidino-2-phenylindole (DAPI). After extensive washing with PBS, coverslips were mounted onto slides.

For GR immunofluorescence analysis, DPSCs were plated in glass chamber-slides and cultured until they reached 80% confluence. Cells were rinsed twice with PBS, and fixed with 4% paraformaldehyde in PBS for 10 min at RT. The following steps were carried out as previously described. The rabbit polyclonal primary antibody to GR was purchased from Thermofisher (1:1000). This step was followed by incubation with the secondary antibody FITC-conjugate (Abcam). Nuclear counterstaining was performed with 4,6-diamidino-2-phenylindole (DAPI).

### 2.9. Transfection (shRNA Construct, Transfection, and Stable Selection)

shHDAC2, and negative control short hairpin RNA (shRNA) (mock) neomycin-resistant SureSilencing shRNA plasmids were purchased from SABioscience (Qiagen, Milano, Italy). DPSC cells were transfected with Lipofectamine 3000 (Invitrogen), according to the manufacturer’s instructions. Clones with downregulated expression of HDAC2 were selected with 500 µg/mL G418 (GIBCO).

### 2.10. Gene Expression Analysis

Gene expression was detected by using the Applied Biosystems^®^ TaqMan^®^ Array Plate for: human osteogenesis (Cat. No 4414096, Applied Biosystems, Foster city, CA, USA). The panel of genes in these plates contains 92 Taqman assays for genes that are associated with the correspondent biological process, and four Taqman assays of endogenous control genes. Samples were analyzed by using the ViiA 7 Real-Time PCR System (Thermo Fischer, Rodano, Milan,). The results were analyzed by using the DataAssist™ Software, version 3 (supplied by Applied Biosystems, Foster city, CA, USA), which enables a rapid and comprehensive interpretation of TaqMan^®^ Array Plate results. DataAssist™ Software provides a filtering procedure for outlier removal, various normalization methods based on single or multiple genes, and relative quantification (RQ) analysis of gene expression through a combination of statistical analysis and interactive visualization. DataAssist™ Software is freely available at http://www.appliedbiosystems.com/DataAssist.

The expression of genes not present in the assay was performed by qRT-PCR.

### 2.11. Protein Extraction and Co-Immunoprecipitation

Cell lysis and the extraction of separate cytoplasmic and nuclear protein fractions was performed with NE-PER™ Nuclear and Cytoplasmic Extraction Reagents (Thermo Fisher).

Co-immunoprecipitation of cytoplasmic proteins was performed with Pierce Protein A/G Magnetic Beads (Thermo Fischer) following the manufacturer’s instructions.

Equal amounts of protein were fractionated by Sodium Dodecyl Sulphate - PolyAcrylamide Gel Electrophoresis (SDS-PAGE), and subsequently transferred onto nitrocellulose membrane. Immunoblots were visualized using Supersignal^®^ West Pico Chemiluminescent substrate (Thermo Scientific, Rockford, USA). Proteins were detected with: anti-GR (1:100, Thermo Fischer Scientific), and anti-HDAC2 (1:100, Abcam).

The protein expression of the tested protein was quantitatively analyzed by the ratio of the gray value between the target protein and GAPDH (1:5000, Abcam), for cytoplasmic proteins, and Lamin (1:1000, Abcam), for nuclear proteins, using Image J software.

### 2.12. Chromatin Immunoprecipitation

Quantitative Chromatine Immunoprecipitation (ChIP) analysis was performed with the Pierce Agarose ChIP Kit (Pierce Biotechnology, IL). All reagents were prepared as described by the manufacturer. Briefly, samples were cross- linked with 1.0% formaldehyde. Anti-GR antibody (Thermo Fischer Scientific) was used with Protein A/G plus Agarose to adsorb immune-specific complexes. Normal rabbit IgG was used as the negative control. Finally, purified DNA was analyzed by real-time PCR (Applied Biosystems, Monza, Italy) using appropriate primers. ChIP-qPCR data were normalized by using the Fold Enrichment Method. Each sample was analyzed in triplicate. Each experiment was performed on at least three separate occasions.

The following primers were used to amplify segments that overlap with the appropriate regions:

*nGRE1* fw: CAGGACATGTCTTCCCTCTCT, *nGRE1* rev: CCCGTAGTGCTCCGATAAGTA

mapping at about −800 bp.

*nGRE2* fw: TAAACAGTGCTGGAGGCTGG, *nGRE2* rev: GTGAGGGGCTCTCATGGTGTC

overlapping the TATA box.

### 2.13. Bioinformatic Analysis

The web site Gene Promoter Miner (GPMiner) is used to search for transcriptional regulatory elements in mammalian gene promoter regions. In this study, GPMiner was used to identify the glucocorticoid receptor element (namely nGRE1) on the osteocalcin promoter.

### 2.14. Flow Cytometry

Cells were detached using trypsin-EDTA (200 mg/L EDTA, 500 mg/L trypsin; Cambrex, East Rutherford, NJ, USA). Intracellular staining for OC was performed using the FIX & PERM Cell Fixation & Cell Permeabilization Kit (Life Technologies Laboratories, Monza, Italy, http://www.lifetechnologies.com) according to the manufacturer’s procedure. At least 300,000 cells were incubated with direct fluorescent-conjugated antibody. The primary antibody used was: Phycoerythrin (PE)-conjugated anti-h-OC (BD Biosciences). For negative controls, cells were stained with an isotype control antibody. Labeled cells were analyzed by flow cytometry, using a FACS Aria III (BD Biosciences, San Jose, CA, USA), and all data were analyzed with FCS Express version 3 software.

For cell-cycle analysis, cells were detached by trypsinization, and then fixed with ice-cold 80% ethanol. The cells were centrifuged and then stained with a solution of 50 µg/mL propidium iodide and 80 µg/mL RNase A for 60 min at 4 °C in the dark. DNA content and cell cycle distribution were measured with a FACSARIA III. All of the data were analyzed with FCS Express version 3 software.

### 2.15. Statistical Analysis

All experiments were carried out in triplicate and repeated at least three times. The results were presented as means ± SEM with *p* ≤ 0.05 being considered statistically significant. Western blot images were semi-quantitatively analyzed with ImageJ software (NIH, University of Wisconsin, Madison, WI, USA). Statistical analysis was performed by an analysis of variance (ANOVA) plus Bonferroni’s *t*-test. A *p*-value of <0.05 was considered to indicate a statistically significant result.

## 3. Results

### 3.1. HDAC Inhibitors Enhance the Expression of Osteogenic Markers

Since HDACi causes growth arrest and apoptosis in tumor cells, as well as in some non-tumor cells, we started by determining the non-cytotoxic doses of VPA, MS275, TSA, and SAHA for DPSCs, by MTT and cell cycle analyses. DPSCs were cultured with different concentrations of HDACi for 24 and 48 hours. We did not detect appreciable differences in cell growth for up to 48 hours of treatment, and up to a concentration of 1.5 mM for VPA, 2.5 µM for MS275, 100 nM for TSA, and 1 µM for SAHA. We then determined the effects of HDACi on the cell cycle by flow cytometry. No significant changes in cell cycle profiles were observed in DPSCs that were cultured with the above-mentioned concentrations of HDACi for 48 hours, with respect to the control. In fact, regardless of treatment, the cells were better distributed in the G_0_G_1_ phase (Appendix A).

Once it was established that there was no cytotoxicity in terms of growth arrest and blockage of the cell cycle, we determined the effects of VPA, MS-275, TSA, and SAHA on osteogenic marker expression of DPSCs. DPSCs cultured in basal medium were considered as controls. RT-qPCR analysis revealed that all HDACi increased the mRNA levels of OPN and BSP, which are osteogenic markers that are involved in matrix deposition. Interestingly, the results showed also that VPA and MS275 led to a decrease in OC mRNA levels, whereas SAHA treatment induced an up-regulation (Figure 1).

Although at the molecular level, osteocalcin expression decreased with VPA and MS275, alizarin red staining showed that they induced a higher production of calcification nuclei, compared to other substances (Appendix A) indicating that these two HDACi could promote osteogenic differentiation.

### 3.2. Bone Formation In Vivo

To evaluate the effects of different HDAC inhibitors in vivo, we performed a transplantation assay in NOD/SCID mice. To this aim, DPSCs treated with the HDACi for 48 hours were placed on a collagen scaffold (Gingistat^®^) and induced to differentiate in osteogenic medium for 15 days.

Constructs of untreated cells seeded on the scaffold and cultured in basal medium and induced in osteogenic medium were used as negative and positive controls, respectively. At the end of 15 days, the scaffold-/-cell constructs were subcutaneously transplanted into NOD / SCID mice and explanted after 60 days. The specimens were subjected to Hematoxylin and Eosin (H&E) and Alizarin red staining (Figure 2A,B). H&E staining highlighted a strong tendency of VPA-treated cells to form dense connective tissue (Figure 2A). Within this tissue a rather large portion that was more dense and organized was visible and highly stained (very eosinophilic), similar to a bone ossification center and covering most of the construct’s cross-section. This bone tissue generated within the portion was already remodeled and lamellar. Other smaller bone formations were visible peripherally. In the samples treated with MS-275, a comparable condition, but with a smaller bone formation center, was observed. TSA-treated DPSCs formed an organized tissue that was similar to the differentiated sample, whereas SAHA-treated cells displayed no organization, as well as in negative control. Indeed, a 3D construct with DPSCs cultured in basal medium showed less organized tissue structure with a lower cell density; notably, the scaffold meshes were still visible (poorly degraded). In the 3D construct with DPSCs cultured in osteogenic medium, only dense but poorly organized tissue was observed, with few early ossification spots. Alizarin Red staining confirmed bone tissue formation in the VPA and MS-275 samples, but not in the other ones (Figure 2B). In fact, in these samples, mineralization deposits were observed, in correspondence with organized bone structures, as highlighted by H&E staining (Figure 2B), and in confirmation with the in vitro experiments.

Sections were also subjected to immunofluorescence analysis to evaluate the expression of OPN and OC (Figure 3A,B). In the samples treated with VPA, there was a widespread positivity of OPN in the tissue portion, which also resulted in positive results according to Alizarin red, while in the surrounding tissue, this marker was absent (Figure 3A). In the constructs treated with MS275, TSA and SAHA, OPN expression was weak and did not highlight the presence of specific bone structures.

OC staining, on the other hand, was homogeneously distributed in all sections except for those treated with SAHA, in which osteocalcin expression was negative. In the VPA treated-sample, OC had a more peripheral distribution, but this was always inside the bone-like tissue that was newly formed (Figure 3B).

The results obtained show that although osteocalcin is less well-expressed in the samples treated with VPA, bone formation is not compromised; on the contrary, it seems to be more organized than its differentiated control. Consequently, our findings led us to focus on the effects of VPA on DPSCs during osteogenic differentiation, and in particular on the molecular mechanisms underlining the reduction of osteocalcin expression.

### 3.3. HDAC2 and Glucocorticoid Receptor Involvement in VPA-Treated DPSCs

In our previous study, we demonstrated that the reduction of OC expression by VPA is mediated through HDAC2 inhibition [33]. Therefore, we decided to investigate how HDAC2 could be involved in such downregulation.

For this purpose, we performed a knockdown for HDAC2 in DPSCs and evaluated the expression of a series of genes and transcription factors using Taqman Arrays. The results showed that, other than changes in genes directly related to osteogenic differentiation, variations in numerous transcription factors was detectable (Appendix A).

After carrying out a study of the literature on the link between these factors and the expression of OC, we identified the glucocorticoid receptor (GR) that was possibly responsible for OC downregulation [37,38,39]. Therefore we performed, first of all, a qRT-PCR analysis, and we demonstrated that GR expression in VPA-treated DPSCs and in those silenced was upregulated, with a statically significant increase (Figure 4A). Then, we performed an immunofluorescence assay, by which we confirmed molecular data showing an increase of GR in VPA-treated and shHDAC2 cells. Surprisingly, IF analysis revealed not only a greater expression for GR, but also larger localization and distribution in the nucleus than in the cytoplasm (Figure 4B).

In order to confirm and quantify what was observed for immunofluorescence, we performed a Western-blot analysis for cytoplasmic and nuclear fractions, and we showed that the expression of GR was higher in treated and in silenced cells at the cytoplasmic, as well as at the nuclear level (Figure 5A,B).

To correlate the relation between GR and HDAC2, we performed a co-immunoprecipitation analysis of the cytoplasmic fraction. Western blotting revealed a strong presence of HDAC2 in the control samples immunoprecipitated with GR, but not in VPA-treated or silenced ones (Figure 5C). The presence of HDAC2 in the cytoplasmic immunoprecipitate is a good indication of the formation of a HDAC2–GR complex/interaction that maintains the receptor in the inactive form in the cytoplasm.

### 3.4. GR Binding on the nGRE Sequence at the Osteocalcin Promoter

Many reports indicate that GR can act on the osteocalcin gene by direct *cis*-repression [43,44,45].

To ascertain whether GR negatively interacts with the OC promoter in our models, we carried out a chromatin immunoprecipitation (ChIP) analysis.

The results showed that GR predominantly bound the nGRE sequence that we identified by GPMiner (http://gpminer.mbc.nctu.edu.tw/show_prediction/show.php?OS=human&ID=20180110_193635&scale=3&GC_window=15&TFBS_core=1.00&TFBS_matrix=0.95&OR_zscore=5&OR_number=20&OR_occur=2&stability_window=15&miRNA_MFE=&miRNA_score) in the osteocalcin promoter in VPA-treated DPSCs, and in those that were silenced. The nGRE sequence overlapping the TATA box was only weakly bound (Figure 6).

### 3.5. GR Inhibition by RU-486-Recovered OC Expression in DPSCs Treated with VPA

To confirm that GR was directly involved in osteocalcin downregulation, we incubated cells with RU-486 for 24 hr. The synthetic steroid molecule RU-486 is a GR antagonist, and it was used as a tool to block GR function. In DPSCs treated with VPA, and in those that are HDAC2-silenced, osteocalcin expression was recovered, and it even increased when the cells were incubated with RU-486, as shown by FACS analysis in Figure 7. These results further confirm that OC reduction in treated and silenced cells is due to GR.

## 4. Discussion

Mesenchymal stem cell methodologies have highly impacted modern therapeutic strategies that are aimed to replace damaged cells or tissues. The multipotent stem cell fate is dependent on important transcription, signaling, and epigenetic factors [46]. Among the epigenetic regulators, histone deacetylases (HDAC) have important roles in cell physiology, differentiation, and developmental decisions [22]. Consequently, the use of HDAC inhibitors (HDACi) elicits cell-restricted responses, determining effects in stem cell differentiation pathways. HDACi can reverse the repressing or activating epigenetic traits that characterize genes that are involved in the regulation of self- renewal or differentiation.

Considering that non-histone proteins are also targets for deacetylation by HDAC, it is expected that the analysis of these changes [47,48] in the course of stem cell differentiation will shed light on the functions and applications of HDAC inhibitors. The elucidation of these mechanisms can open up new opportunities in the field of regenerative medicine and stem cell biology, by the development of new chemical compounds that allows a target-based approach.

Moreover, combining HDAC inhibitors with other small molecule effectors and/or biomaterials that mimic a native microenvironment, it is possible to provide valuable tools to improve stem cell applications for tissue regeneration therapies [49,50,51].

In this study, we utilized different HDAC inhibitors on DPSCs to test their effects on bone formation in vivo.

HDACi are classified on the basis of their chemical structures, and they inhibit the enzymatic activity of HDAC with different efficiencies and specificities. Hydroxamic acids, such as TSA and SAHA, are pan-inhibitors; the short-chain fatty acid, VPA, in contrast, selectively inhibits HDACs of class I (HDAC1, 2,3,8); benzamide MS-275, in turn, is selective only for a subclass of HDAC class I (preferentially inhibits HDAC1 and 2 at low doses as we considered [52,53]. Among them, VPA—an FDA-approved short-chain fatty acid—has been widely used for more than 20 years for the treatment of different neurological disorders [54]. Interestingly, patients with epilepsy show an increased risk of fracture. Thus, the effect of antiepileptic drugs on bone turnover has been extensively studied. VPA does not induce hepatic enzyme activity, and it therefore would not be expected to reduce bone mineral density via this mechanism [55,56].

VPA acts as a potent HDAC inhibitor at the concentrations that are used clinically, suggesting that HDACs have a role as targets for therapy in regenerative medicine for skeletal pathologies with bone loss [57,58].

Therefore, since the safety of VPA in patients has been already tested, its clinical use for new molecular approaches is of great interest.

Valproic acid demonstrated a better capability for producing a well-organized bone tissue structure in vivo.

In NOD/SCID mice models, indeed, we obtained a lamellar compact bone tissue structure in VPA-treated constructs, whereas only a dense connective tissue structure in the other constructs was observed. Furthermore, we failed to detect a well-organized tissue in complexes containing DPSCs grown in basal medium, as well as in SAHA-treated samples. Bone tissue formation in VPA-treated samples was confirmed by Alizarin red staining and immunofluorescence analysis for osteogenic markers, which were resultantly concentrated within the organized structure. In addition, it is to be noted that osteocalcin, although poorly expressed, does not compromise bone formation. These results are consistent with previous genetic evidence showing that OC-deficient mice develop a phenotype that is characterized by increased bone formation [59]. Until now, the role of osteocalcin is still unknown. Osteocalcin is an osteoblast-specific gene encoding a secreted protein. Ducy et al. revealed that osteocalcin-deficient mice developed a phenotype that was marked by higher bone mass, and bones of improved functional quality [60]. This could be due to the fact that by inhibiting osteocalcin, which participates only in the post-proliferative osteoblasts stage, these cells can produce more extracellular matrix, and thus, a higher bone mass.

From the results obtained so far, we subsequently decided to focus our studies on the molecular mechanisms that lead to osteocalcin downregulation after VPA treatment. Several molecular mechanisms may be involved in the alteration of OC expression. In a previous paper [33], we demonstrated that the inhibition of OC gene expression by VPA is most likely mediated through HDAC2 inhibition. We decided therefore to silence HDAC2 in dental pulp stem cells, and we evaluated the expression of a series of transcription factors involved in osteocalcin regulation. Thus, we identified GR as being responsible for this downregulation. Pico et al. demonstrated that GR acts on OC genes by direct *cis*-repression [59]. Therefore, we decided to analyze the expression of GR in dental pulp stem cells after treatment with VPA and in shHDAC2 DPSCs. Surprisingly, results obtained by qRT-PCR and Western blotting showed an upregulation of GR, in treated and silenced cells. In particular, protein overexpression was higher at the nuclear level, as shown in Western blotting and in immunofluorescence analyses. The latter demonstrates that both in VPA-treated and in silenced cells, GR translocates into the nucleus, where this factor acts on the osteocalcin promoter, repressing it.

GR is a ubiquitously expressed nuclear receptor that controls the transcription of genes that are critical for metabolism, immunity, development, and responses to stress [61,62]. In the inactive form, it resides in the cytoplasm as a multimeric complex, named the foldosome [36,63]. Post-translational modifications play a crucial role in the regulation of GR activity. Among this, acetylation can affect its stability, subcellular localization, transcriptional activity, and interactions with other proteins. Although enzymes involved in (de-)acetylation events, i.e., histone acetyltransferases and histone deacetylases (HDACs), primarily target the lysines located in histone tails, there is evidence for direct acetylation of other proteins, notably GR.

GR acetylation is ligand-dependent, and it regulates GR function. Before nuclear translocation, GR undergoes acetylations at K494 and K495, located in the hinge region [64]. We hypothesized that HDAC2 helps to keep GR in an inactive state in the cytoplasm, and that VPA treatment allows GR translocation in the nucleus. To this aim, we demonstrated, for the first time, that GR binds HDAC2 in the cytoplasm, and that the lack of this enzyme causes GR nuclear translocation. Indeed, co-immunoprecipitation analysis of the cytoplasmic fraction evidenced that the GR-HDAC2 complex in DPSCs were untreated. In VPA-treated DPSCs, and in those that were HDAC2-silenced, the GR−HDAC2 complex was barely detectable. Until now, it has been known only that the deacetylation of GR by HDAC2 occurred in the nucleus to form a complex that inhibits inflammatory gene expression [65].

GR alters gene expression via several modes of action, including its binding as dimers to GC-responsive elements (GRE) that are present in the promoters of hormone-responsive genes, and the interaction of the monomeric receptor with DNA-bound transcription factors.

In osteoblasts, *OC* is among the direct targets of GR. Early studies suggested that direct binding of GR near the OC gene’s TATA box contributes to transcriptional repression [37,38,39]. Other GR-binding sites were found at the OC box [40,66,67] and promoter domains, which are more distal.

Thus, to demonstrate that GR directly binds to the nGRE consensus sequence at the osteocalcin promoter, we performed ChIP analysis. We found that GR directly binds to the nGRE sequence, mostly at a site about 800 bp from the transcription start site in the osteocalcin promoter, whereas binding at the TATA region was weaker in VPA-treated DPSCs and in those that were silenced. GR involvement in OC downregulation was further ascertained by blocking its activity with RU-486 in DPSCs incubated with VPA.

## 5. Conclusions

In this study, we highlight the molecular mechanisms by which VPA exerts its effects in enhancing bone formation, with a concomitant reduction of osteocalcin expression. We identify, for the first time, the glucocorticoid receptor (GR) as being responsible for this regulation. GR forms a complex with HDAC2 in the cytoplasm of treated cells, and in the absence of this enzyme, it translocates into the nucleus, exerting negative effects on OC transcription.

All the above is of interest, as epigenetic mechanisms are potentially reliable alternatives to the existing strategies for MSCs differentiation. Pretreatment with HADCi for the enhancement of osteogenic differentiation indicates that HDAC inhibitors could be used for in vivo bone engineering.

## Figures and Tables

**Figure 1 cells-08-00217-f001:**
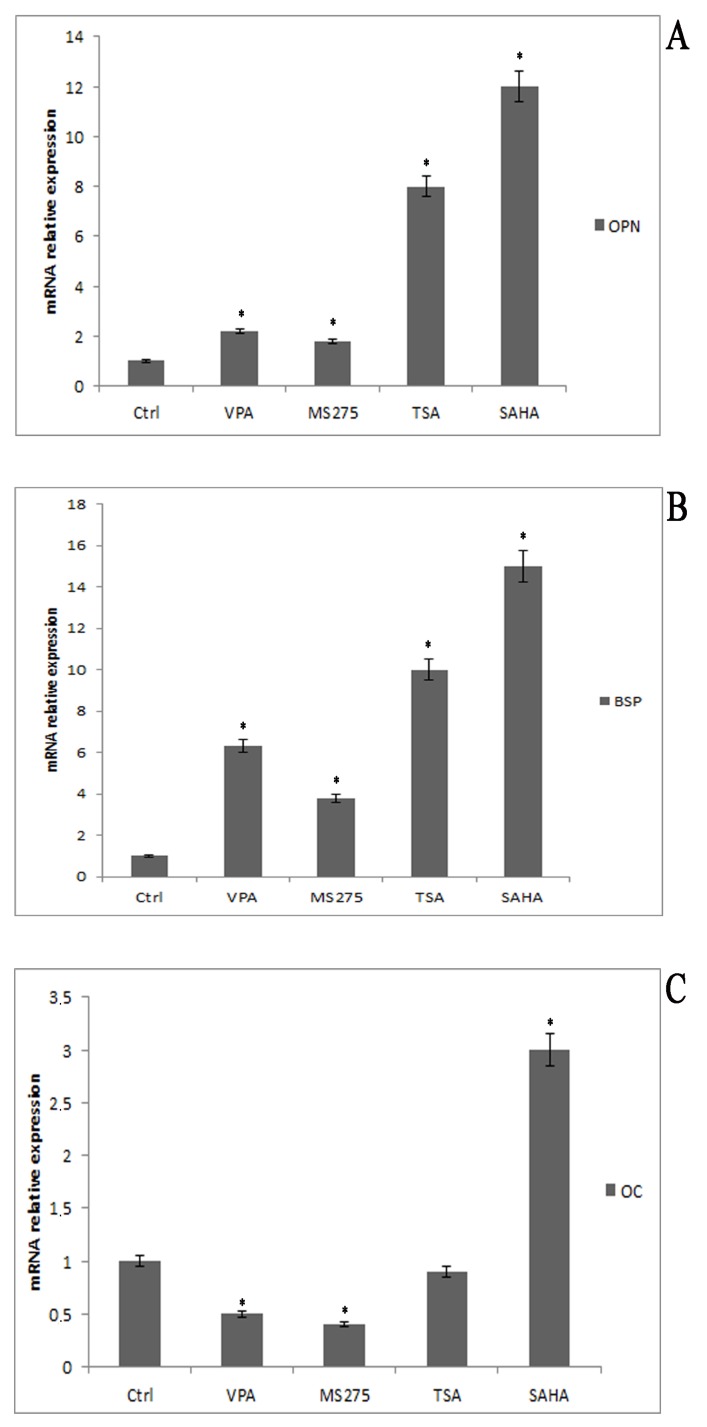
Osteogenic genes expression after treatment with histone deacetylase (HDAC) inhibitors. Dental pulp stem cells (DPSCs) treated with different HDAC inhibitors expressed high levels of osteopontin (OPN) (**A**) and bone sialoprotein (BSP) (**B**). Only in valproic acid (VPA)- and MS-275-treated cells was the osteocalcin (OC) expression level lower (**C**). Data are represented as mean ± SEM. * *p*  <  0.05 compared to control cells.

**Figure 2 cells-08-00217-f002:**
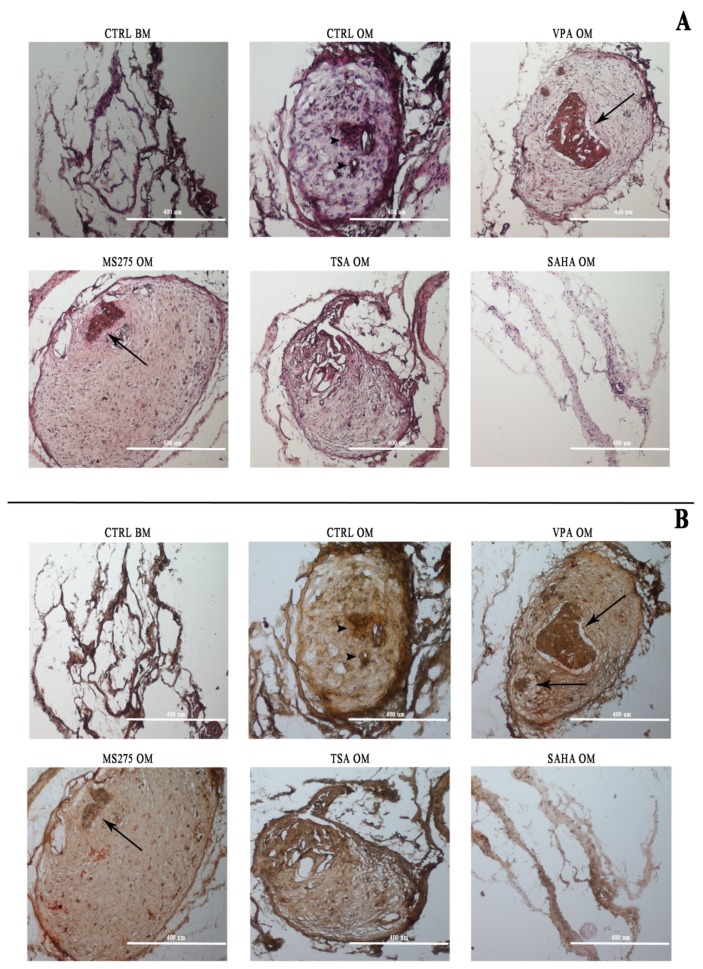
Histological evaluation after in vivo implantation. **A**: H&E staining. H&E staining highlighted a strong tendency for VPA-treated cells to form a dense connective tissue. Within this tissue, a rather large portion is visible that is even more dense and organized, and highly stained, similar to a bone ossification center. Other smaller bone formations are visible (arrows in the VPA and MS275 treated cell images). Most of the bone tissue generated within the portion was already remodeled and lamellar. In the 3D construct with the DPSCs cultured in osteogenic medium, only dense but poorly organized tissue was observed, with few early ossification spots (arrowheads in CTRL OM). **B**: Alizarin Red staining confirmed a bone structure in the VPA samples. Scale bar: 400 μm.

**Figure 3 cells-08-00217-f003:**
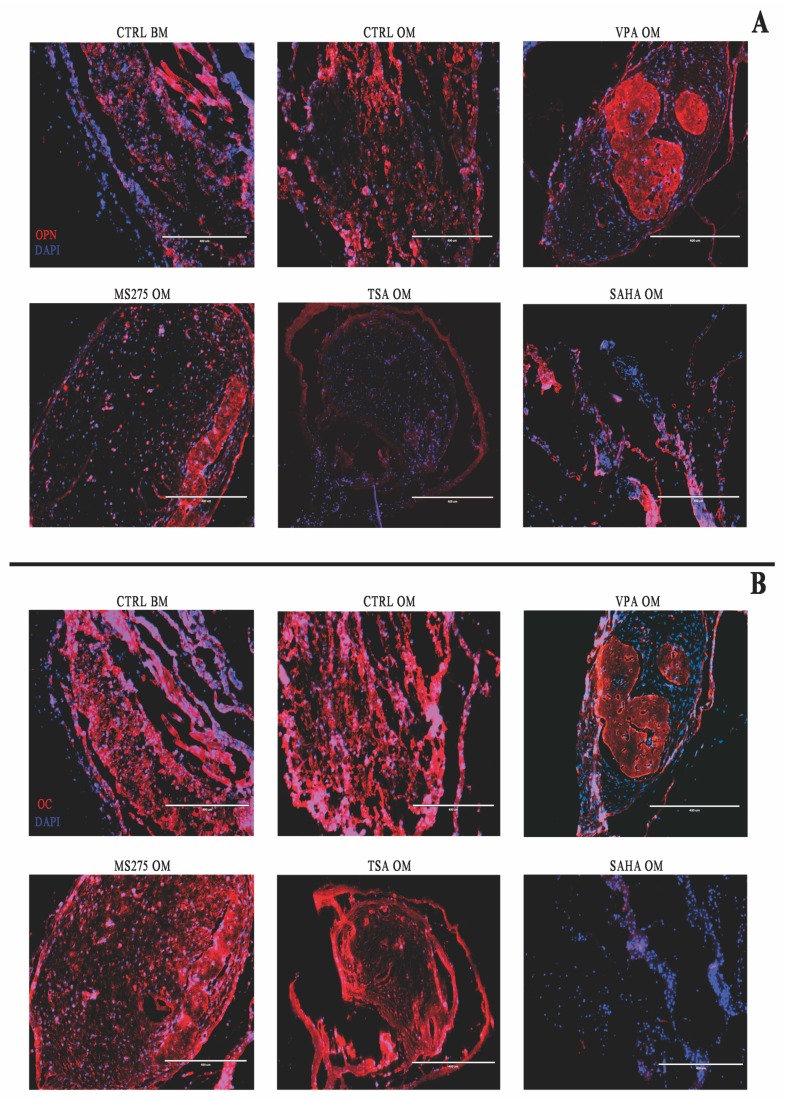
Immunofluorescence analysis after in vivo implantation. Immunofluorescence analysis evaluating the expression of OPN (**A**) and OC (**B**): OPN positivity was present only within the bone structure in the samples treated with VPA, while in the surrounding tissue, this marker was virtually absent. OC staining, on the other hand, has a more peripheral distribution, but it is always inside the bone-like tissue, even if it is barely detectable with respect to the control samples. Scale bar: 400 µm.

**Figure 4 cells-08-00217-f004:**
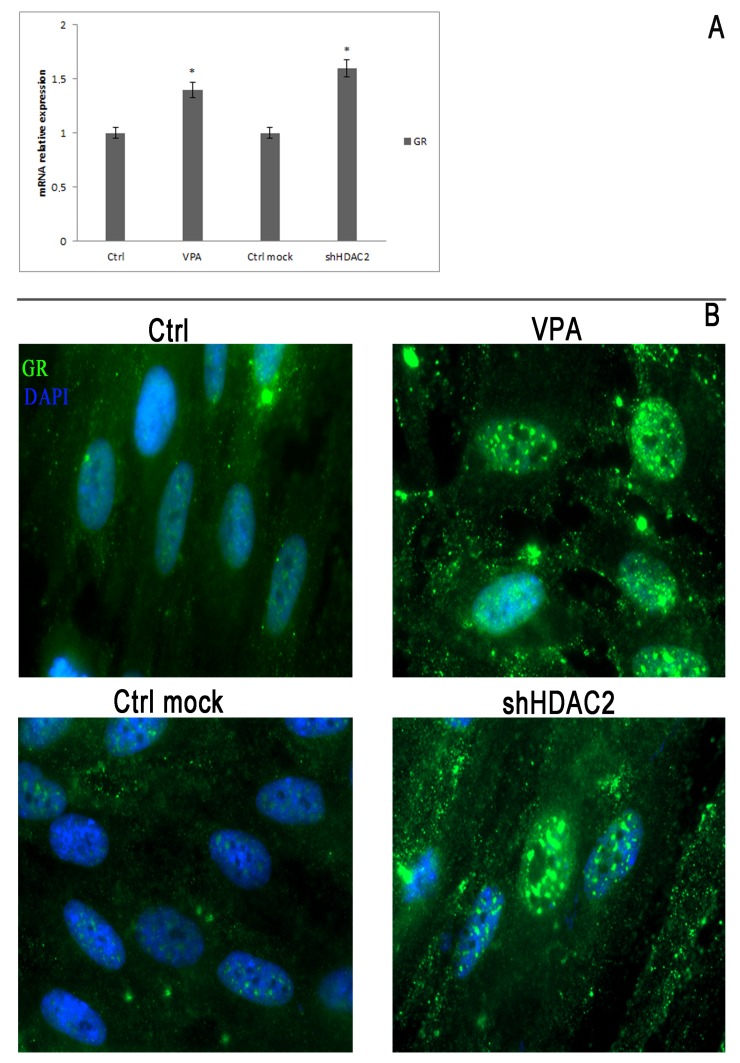
GR expression and immunofluorescence analysis. **A**: qRT-PCR of the glucocorticoid receptor in cells treated with VPA, and in cells silenced for HDAC2. The expression of GR in cells treated with VPA and in shHDAC2 cells was significantly higher, with respect to the control. Data are represented as mean ± SEM. * *p*  <  0.05 compared to control cells. **B**: Immunofluorescence analysis for GR. IF analysis revealed more concentrated staining in the nuclei of treated or silenced cells with respect to the controls. Magnification: 100×.

**Figure 5 cells-08-00217-f005:**
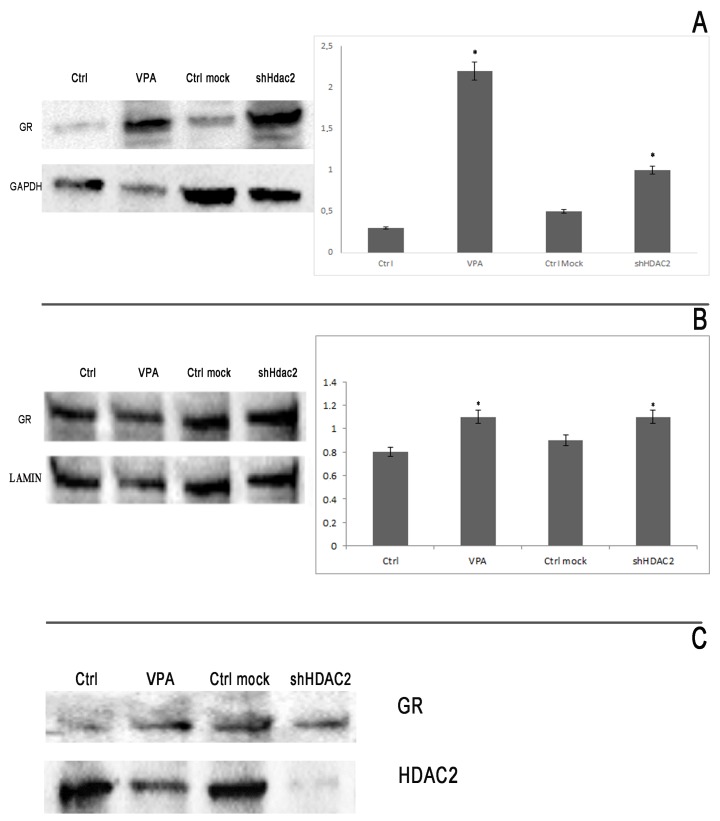
Western blotting of cytoplasmic and nuclear proteins. GR is overexpressed in cytoplasmic fraction in VPA-treated and in silenced cells (**A**), as well as at the nuclear level (**B**). Histograms represent a densitometric analysis by ImageJ. * *p*  <  0.05 compared to control cells. (**C**). Co-IP for GR and HDAC2. The abundance of HDAC2 in the control samples is a clear signal of the formation of the GR/HDAC2 complex that maintains GR in an inactive form in the cytoplasm.

**Figure 6 cells-08-00217-f006:**
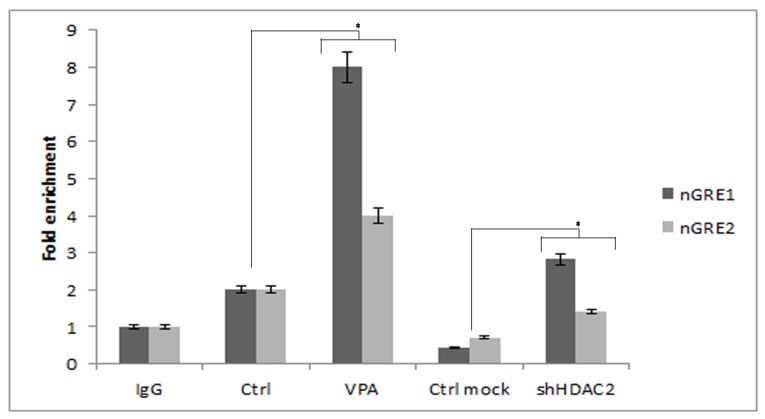
ChIP analysis of GR in the osteocalcin promoter. GR binds predominantly on the nGRE1 sequence in the osteocalcin promoter in VPA-treated DPSCs, and in those that are silenced. The nGRE sequence near the TATA box (nGRE2) was only weakly bound. Data are represented as mean ± SEM. * *p*  < 0.05 compared to the control.

**Figure 7 cells-08-00217-f007:**
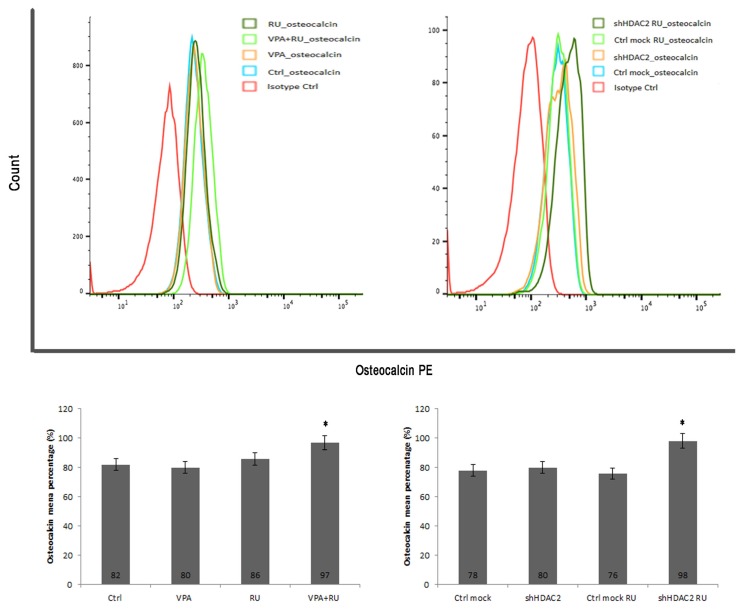
FACS analysis of the RU-486 effect on OC expression. In DPSCs treated with VPA, and in those that are HDAC2- silenced, osteocalcin expression was recovered, and even increased when the cells were incubated with RU-486, a GR antagonist. * *p* <  0.05 compared to control.

**Table 1 cells-08-00217-t001:** Primers sequences for quantitative Real-time Polymerase Chain Reaction (qRT-PCR).

GENE	Accession Number	Sequence	Amplicon Size (bp)	Tm (°C)	Ta (°C)
*GAPDH*	*NM_002046.7*	*FW: ggagtcaacggatttggtcg* *REV: cttcccgttctcagccttga*	180	81.7	60
*OC*	*NM_199173.6*	*FW: ctcacactcctcgccctattg* *REV: cttggacacaaaggctgcac*	108	87.51	60
*OPN*	*NM_001040058.2*	*FW: gccgaggtgatagtgtggtt* *REV: tgaggtgatgtcctcgtctg*	101	79.33	58
*BSP*	*NM_004967.4*	*FW: ctggcacagggtatacagggttag* *REV: actggtgccgtttatgccttg*	182	79.46	60
*GR*	*NM_000176.3*	*FW: ggagaggggagatgtgatgg* *REV: agggtgaagacgcagaaacc*	72	79.16	60

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
