# Peer review of "Cytoplasmic Interactions between the Glucocorticoid Receptor and HDAC2 Regulate Osteocalcin Expression in VPA-Treated MSCs"

_cells, 2019, doi:10.3390/cells8030217_

Round 1
Reviewer 1 Report
The manuscript entitled “Cytoplasmic Interaction between Glucocorticoid Receptor and HDAC2
regulates Osteocalcin expression in VPA-treated MSCs” reports results om the use of HDAC inhibitor to promote the differentiation of the MSCs into osteogenic lineage. The authors also found that there is interaction between the glucocorticoid receptor and the HDAC in the cytoplasm, which is reported for the first time in this manuscript.
Abstract: This is quite vague, more details and more specific information is necessary in the abstract. The authors should mention which in vivo model and where the cells were transplanted in brief. The abstract in the current form does not give the gist of the study and rises more question. The abstract needs to be re-written.
Page 2 – Line 64 – Please briefly explain what HDAC inhibitors are and their role and importance in gene expressions well as in MSCs differentiation.
Is the mechanism associated with the GR is the same for all the MSCs differentiation process irrespective of their source? Or is this source-dependent? Is use VPA or HDACi, the only method for MSCs differentiation into osteoblast? How is this method would be more advantageous compared to other methods of differentiation?
Line 101 – Please mention the gender of the adults participated in the study.
Line 115 – Once the cells were plated, how were they selected and characterized?
Line 117 – What was the passage when the cells were differentiated and when the qPCR and WB analysis was done? Please mention the concentration and the time points at which the HDACi were added to the cells. These should be mentioned in the method section rather than in the result section.
Line 137 – in vivo grafting should be re-written. In the current form, this is very confusing as to where exactly the transplantations were done. How many mice were used in the study? Also, were the cells pre-stained with a cellular marker?
Line 160 – what/which tissue was collected. This is rather confusing.
Line 171 – what is the concentrations of the antibody used for both histology and Western blot?
Result section: The result section starts with the methodology, though this might be helpful for the readers, more details should go into the methods section rather than the result section.
Conclusion: This is a good attempt by the authors to look at the effects of HDACi, GR interaction and the differentiation of MSCs into osteoblasts. More descriptions about the HDACi, VP and their importance should be discussed. Also, the authors should re-write the animal section as this is rather confusing about the transplants and what type of tissue was analyzed. The methods and result section should be more organized as “in vitro” and “in vivo” for the better flow for the readers. Overall, the authors should mention the novelty of the study, and currently this study should be made more interesting for the readers. This manuscript needs major revisions.
Author Response
February 6th, 2019
Dear Editor-in-Chief,
enclosed please find the revised form of the manuscript entitled “Cytoplasmic Interaction between Glucocorticoid Receptor and HDAC2 regulates Osteocalcin expression in VPA-treated MSCs”.
We have answered to each point raised by the referees #1, #2 and #3. The detailed answers are attached below this letter in red and, also in the manuscript, all corrections are in red.
Therefore, having answered to all the points, we hope that the manuscript, in its present revised form, will be now suitable for publication in “Cells”.
Yours faithfully,
With best regards,
Gianpaolo Papaccio Prof. MD., PhD
Reviewer #1:
We thank the reviewer# 1 for the constructive and helpful suggestions.
We have provided our point-by-point responses to the reviewer’s comments:
1. Abstract: This is quite vague, more details and more specific information is necessary in the abstract. The authors should mention which in vivo model and where the cells were transplanted in brief. The abstract in the current form does not give the gist of the study and rises more question. The abstract needs to be re-written.
R. We have re-written the abstract following the reviewer’s suggestions, as follows:
Epigenetic regulation has been considered as an important mechanism to influence stem cell differentiation. In particular, histone deacetylases (HDACs) have been shown to play a role in osteoblast differentiation of Mesenchymal Stem Cells (MSCs). In this study, the effect of the HDAC inhibitor, Valproic Acid (VPA), on bone formation in vivoby MSCs was determined. Surprisingly, VPA treatment, unlike other HDAC inhibitors, produced a well-organized lamellar bone tissue when MSCs - collagen sponge constructs were implanted subcutaneously into NOD/SCID mice, although a decrease of osteocalcin (OC) expression was observed. Consequently, we decided to investigate the molecular mechanisms by which VPA exerts such effects on MSCs. We identified the Glucocorticoid receptor (GR) as responsible for that downregulation and suggested a correlation between GR and HDAC2 inhibition after VPA treatment, as evidenced by HDAC2 knockdown.
Furthermore, using co-immunoprecipitation analysis, we demonstrated for the first time, a binding between GR and HDAC2 in the cytoplasm. Additionally, Chromatin immunoprecipitation (ChIP) assay confirmed the role of GR in OC downregulation showing a recruitment of GR to the nGRE element in the OCpromoter. In conclusion, our results highlight the existence of a cross-talk between the GR and HDAC2 providing a mechanistic explanation of the influence of the HDAC inhibitor (namely VPA) on osteogenic differentiation in MSCs.
Our findings open new directions in targeted therapies and offer new insights into the regulation of MSCs fate determination.
2.Page2-line64-Please briefly explain what HDAC inhibitors are and their role and importance in gene expressions well as in MSCs differentiation.
R. We have better explained the role of HDAC inhibitors and their importance in gene expression and MSCs differentiation, as follows:
The regulation of mechanisms underlining the stem cell state and/or differentiation is essential for the development of stem cell-based therapy. The events that govern stem cell differentiation are predominantly epigenetics. Indeed, specific patterns of DNA methylation and histone modifications play an important role in the induction of MSC differentiation toward specific lineages. Histone acetylation is the most well-studied histone modification and has been shown to be an important process of gene regulation. Histone acetylation contributes to the formation of a more relaxed and thus transcriptionally active chromatin structure. In contrast, histone deacetylation is associated with a condensed chromatin leading to transcriptional repression. The level of cellular histone acetylation is regulated by the opposing activities of histone acetyltransferases (HAT) and histone deacetylases (HDACs), which reversibly catalyze the acetylation and deacetylation mechanisms, respectively [20, 21]. HDACs have been shown to regulate several important biological processes, including cell proliferation, differentiation, and development, by forming complexes with various transcription factors and transcriptional coregulators [22]. The inhibition of HDAC activity typically leads to de-repression of transcription. HDAC inhibitors (HDACis) are natural or synthetic small molecules that can inhibit the activities of HDACs and promote efficient and temporally regulated control of gene expression. HDACis have shown very potent effects in stem cell differentiation pathways. They may promote either self-renewal [23;24] or differentiation [25;26] depending on the stem cell status, the dose employed and the cell type [27]. Therefore, application of epigenetic regulators, such as HDAC inhibitors, may be valuable for stem cell‐based interventions. There is growing evidence that some HDACi also stimulate osteogenic differentiation of MSCs [28-32].
20. Legube, G., Legube, D. Regulating histone acetyltransferases and deacetylases. EMBO Rep.; 2003, 4:944–947.
21. Kuo, M.H., Allis, C.D. Roles of histone acetyltransferases and deacetylases in gene regulation. Bioessays.; 1998, 20:615–626
22. Haberland, M., Montgomery, R.L., Olson, E.N. The many roles of histone deacetylases in development and physiology: Implications for disease and therapy. Nature Reviews Genetics, 2009, 10:32–42
23. Ware, C.B., Wang, L., Mecham, B.H., et al. Histone deacetylase inhibition elicits an evolutionarily conserved self-renewal program in embryonic stem cells. Cell Stem Cell. 2009;4:359–369
24. Hayashi, K., Lopes, S.M.C.D.S., Tang, F., Surani, M.A. Dynamic equilibrium and heterogeneity of mouse pluripotent stem cells with distinct functional and epigenetic states. Cell Stem Cell. 2008;3:391–401
25. McCool, K.W., Xu, X., Singer, D.B., Murdoch, F.E., Fritsch, M.K. The role of histone acetylation in regulating early gene expression patterns during early embryonic stem cell differentiation. Journal of Biological Chemistry. 2007;282:6696–6706.
26. Karantzali, E., Schulz, H., Hummel, O., Hubner, N., Hatzopoulos, A.K., Kretsovali, A. Histone deacetylase inhibition accelerates the early events of stem cell differentiation: transcriptomic and epigenetic analysis. Genome Biology.2008
27. Kretsovali, A., Hadjimichael, C., Charmpilas, N. Histone deacetylase inhibitors in cell pluripotency, differentiation, and reprogramming. Stem Cells Int. 2012;2012:184154.
28. Lee, S., Park, J.R., Seo, M.S., Roh, K.H., Park, S.B., Hwang, J.W., et al. Histone deacetylase inhibitors decrease proliferation potential and multilineage differentiation capability of human mesenchymal stem cells. Cell Prolif. 2009;42:711–20
29. Schroeder, T.M., Westendorf, J,J. Histone deacetylase inhibitors promote osteoblast maturation. J Bone Miner Res. 2005;20:2254–63
30. de Boer, J., Licht, R., Bongers, M., van der Klundert, T., Arends, R., van Blitterswijk, C. Inhibition of histone acetylation as a tool in bone tissue engineering. Tissue Eng. 2006;12:2927–37
31. Cho HH, Park HT, Kim YJ, Bae YC, Suh KT, Jung JS. Induction of osteogenic differentiation of human mesenchymal stem cells by histone deacetylase inhibitors. J Cell Biochem. 2005;96:533–42
32. Lee, H.W., Suh, J.H., Kim, A.Y., Lee, Y.S., Park, S.Y., Kim, J.B. Histone deacetylase 1-mediated histone modification regulates osteoblast differentiation. Mol Endocrinol. 2006;20:2432–43
3. Is the mechanism associated with the GR is the same for all the MSCs differentiation process irrespective of their source? Or is this source-dependent? Is use VPA or HDACi, the only method for MSCs differentiation into osteoblast? How is this method would be more advantageous compared to other methods of differentiation?
R. In our previous study we demonstrated that the osteocalcin inhibition after VPA treatment occurred in MSCs derived from dental pulp (DPSCs) as well as in primary osteoblasts and osteoblast-like cells derived from osteosarcoma (Saos-2).[Paino et al.,2014].
Osteoblasts differentiate from bone- specific lineage committed mesenchymal stromal cells (MSCs). Bone lineage commitment is driven by the expression of the transcription factor RUNX2 in MSCs, which further promotes expression of different osteogenic markers in a sequential manner. In in-vitro culture systems, the differentiation of MSCs from different sources into osteoblasts is induced by dexamethasone (Dex), a potent synthetic form of the steroid glucocorticoid (GC). The main downstream effector of GCs is the glucocorticoid receptor (GR), a ligand-inducible transcription factor belonging to the nuclear receptor superfamily. To date, there are different studies suggesting osteogenic differentiation on MSCs in response to glucocorticoids and GR involvement in osteocalcin expression, but no one have demonstrated the close interaction between GR and HDAC2 at cytoplasmic level, as we stated. Moreover, many authors described the GR involvement in Osteocalcin downregulation in mouse and human system (N Leclerc, 2005; Meyer, 1997).
The differentiation of MSCs into osteoblasts may be induced by several factors (osteogenic medium, growth factors, etc.). Modern strategies of tissue engineering and regenerative medicine (TERM) utilize tissue scaffolds, healing promotive factors (e.g., growth factors such as BMP-2), combined with MSCs to improve bone differentiation and/or regeneration but none has achieved a really successful result in bone formation.
We have considered epigenetic study as a valid and possible alternative to these strategies. Moreover, our study allowed to evaluate the molecular mechanisms that may be involved in osteogenic differentiation. Knowing the molecular mechanism underlying osteogenic differentiation is necessary to determine the outcome of cell therapies to be used in the treatment of diseases without resorting to the use of less specific growth factors.
4.Line 101: – Please mention the gender of the adults participated in the study.
R. The gender that we considered is both male and female (50% and 50%), we added this information in line 123.
The gender of the patients has not been evaluated because, on the basis of our experience and literature, it is not a variable that influences the osteogenic differentiation of the dental pulp cells.
5.Line 115: – Once the cells were plated, how were they selected and characterized?
R. We added this part in the manuscript.
Line 138: DPSCs were selected and characterized as previously described (La Noce et al, 2014). Briefly, flow cytometry analyses were performed on hDPSCs at first passage of culture (approximately 1x106cells). Human DPSCs were sorted for CD34 and CD90 positive markers using a FACS Aria III BD (BD Biosciences, Milan, Italy). The purity of sorting was approximately 90%. For phenotypiccharacterization, cells were incubated with FITC-conjugated anti-CD90, PerCP-Cy5.5-conjugated anti-CD105, APC-Cy7-conjugated anti-CD45 (all purchased from BD Pharmingen), and PE-conjugated anti-CD34 (Miltenyi Biotech) and FITC-conjugated anti-bone sialo-protein (BSP) (Biorbyt), anti-CFS-conjugated anti-osteopontin (OPN) (R&D Systems) for evaluation of osteogenic differentiation. As negative controls, cells were stained with an isotype control antibody.
[43] La Noce et al: Dental pulp stem cells: State of the art and suggestions for a true translation of research into therapy. J Dent. 2014
6.Line 117:What was the passage when the cells were differentiated and when the qPCR and WB analysis was done? Please mention the concentration and the time points at which the HDACi were added to the cells. These should be mentioned in the method section rather than in the result section.
R. We have better specified the passage of culture and the concentration of HDACis as follows:
Line 148: For Osteogenic differentiation, whencells at the third passage of culture reached 60–70 % confluency, were induced using osteoinduction medium
Line 155: Cells were treated with 1,5 mM of VPA, 2,5 µM of MS275, 100 nM of TSA and 1 µM of SAHA for 48 hours.
7.Line 137: – in vivo grafting should be re-written. In the current form, this is very confusing as to where exactly the transplantations were done. How many mice were used in the study? Also, were the cells pre-stained with a cellular marker?
R. We have re-written the paragraph following the reviewer’s suggestions.
Line 171: In order to achieve 3D tissue formation, treated and untreated cells were seeded on a Gingistat® (GABA VEBAS) scaffold. This scaffold is a lyophilized collagen type I sponge that was cut under sterile conditions into 5 × 5 × 5 mm cubes. Scaffold cubes were placed in six-well plates and a cell suspension of 1 × 106cells contained in 200 µl medium was pipetted onto the top of each cube. Cells were allowed to adhere under a humidified atmosphere at 37°C and 5% CO2for 4 hours. The seeded scaffolds were then placed in tubes containing osteogenic medium for 15 days. Osteogenic medium was changed twice a week.
Twenty-four female NOD/SCID mice, 6-week-old, were purchased from Charles River (Charles River Laboratories International, Inc, Milan, Italy) and acclimatized for a week prior to experimentation. The seeded scaffolds (constructs) were implanted into subcutaneous dorsal pockets of the immunodeficient mice. In particular, four mice per group were transplanted with constructs of cells treated with HDACi (VPA, MS275, TSA, SAHA) seeded on the scaffold; cells seeded on the scaffold, but cultured in basal medium, and cells without HDACi treatment, but induced in osteogenic medium, were used as negative and positive controls, respectively. After 60 days, mice were sacrificed and constructs were removed, fixed in buffered formalin and subsequently analyzed by immunohistochemistry.
8. Line 160 – what/which tissue was collected. This is rather confusing.
R: We modified the sentence as follows:
Line 192: Constructs removed from the subcutaneous excision were fixed in 4% paraformaldehyde…
9. Line 171 – what is the concentrations of the antibody used for both histology and Western blot?
R. We added the dilution of all antibodies used in Material and methods section.
10.Conclusion:
R. We improved conclusion as suggested by this reviewer.
Line 497: In our study we have demonstrated that VPA treatment on DPSCs is able to produce a well-organized bone tissue in vivo, although OC expression decreases. In addition, we have investigated the mechanism on how VPA exerts its effect enhancing bone formation, with simultaneous reduction of osteocalcin expression. We were capable to identify the glucocorticoid receptor (GR) as responsible for this regulation. Therefore, we have demonstrated, for the first time, that GR forms a complex with HDAC2 in the cytoplasm of treated cells, and that in the absence of this enzyme it translocates in the nucleus exerting its negative effect on OC transcription.
All the above is of paramount importance because we have emphasized that epigenetic mechanisms must be considered as a reliable alternative to the existing strategies for MSCs differentiation. Therefore, with our study we were capable to evaluate the molecular mechanisms involved in osteogenic differentiation; to better understand such mechanisms is required to establish the outcome of cell therapies to be used in the treatment of diseases without the use of less specific growth factors.

Reviewer 2 Report
This paper reports a significant contribution to the existing literature on the topic of bone tissue repairing and regeneration. I would suggest acceptance, even if I feel it would be useful to slightly improve conclusions, with the aim to suggest clinical implications related to the findings reported by authors.
Author Response
February 6th, 2019
Dear Editor-in-Chief,
enclosed please find the revised form of the manuscript entitled “Cytoplasmic Interaction between Glucocorticoid Receptor and HDAC2 regulates Osteocalcin expression in VPA-treated MSCs”.
We have answered to each point raised by the referees #1, #2 and #3. The detailed answers are attached below this letter in red and, also in the manuscript, all corrections are in red.
Therefore, having answered to all the points, we hope that the manuscript, in its present revised form, will be now suitable for publication in “Cells”.
Yours faithfully,
With best regards,
Gianpaolo Papaccio Prof. MD., PhD
Reviewer #2
We thank the reviewer # 2 for the constructive and helpful suggestions.
1.This paper reports a significant contribution to the existing literature on the topic of bone tissue repairing and regeneration. I would suggest acceptance, even if I feel it would be useful to slightly improve conclusions, with the aim to suggest clinical implications related to the findings reported by authors.
R. We have improved the conclusions, as suggested by the reviewer. All the amendments are highlighted in red in the revised manuscript.
Line497: In our study we have demonstrated that VPA treatment on DPSCs is able to produce a well-organized bone tissue in vivo, although OC expression decreases. In addition, we have investigated the mechanism on how VPA exerts its effect enhancing bone formation, with simultaneous reduction of osteocalcin expression. We were capable to identify the glucocorticoid receptor (GR) as responsible for this regulation. Therefore, we have demonstrated, for the first time, that GR forms a complex with HDAC2 in the cytoplasm of treated cells, and that in the absence of this enzyme it translocates in the nucleus exerting its negative effect on OC transcription.
All the above is of paramount importance because we have emphasized that epigenetic mechanisms must be considered as a reliable alternative to the existing strategies for MSCs differentiation. Therefore, with our study we were capable to evaluate the molecular mechanisms involved in osteogenic differentiation; to better understand such mechanisms is required to establish the outcome of cell therapies to be used in the treatment of diseases without the use of less specific growth factors.

Reviewer 3 Report
The manuscript “Cytoplasmic Interaction between Glucocorticoid Receptor and HDAC2 regulates Osteocalcin expression in VPA-treated MSCs” from Marcella La Noce et al. describes the effect of valproic acid (VPA) on osteocalcin (OC) expression in dental pulp stem cells (DPCSs). The study gives valuable information on possible regulatory role of glucocorticoid receptor (GR) in this mechanism.
Before considering publication some questions remain to be addressed:
1. The link to HDAC2 is not clearly shown, as the used inhibitors should all inhibit HDAC2 activity.
2. Furthermore, OC expression was investigated as VPA treated cell-constructs showed very good bone structure after subcutaneous culture – despite low OC levels. Final proof that increasing OC levels (stated with RU-486) in these cell-constructs gives comparable bone structure is missing.
3. Using qRT-PCR gene identification (NM_ID), amplicon size, melting curves (selectivity of the method), and efficiency (standard curve) should be given. Furthermore, it should be shown that the used house-keeper was evenly expressed – as the observed differences in gene expression are rather low.
4. In Materials and Methods IF staining against anti-human nuclear antigen is noted but not shown in the results section. Also it should be named from which species the AB (mono- or polyclonal) are. Furthermore, background controls (only secondary AB should be shown)
5. It is stated that Student’s T-test was used for statistics – this is only suitable for comparing two groups that are normally distributed. Here more groups are compared thus an ANOVA based test has to be used. Based on the small sample size this test should be non-parametric.
6. In the graphics it is not clear what statistics are shown: meaning of * should be given in the figure legends – but even more important: it should be clarified which data sets the stated * refers to.
7. Regulation of OC in figure 1 cannot be really seen – please consider changing the figure for clarification.
8. It was stated that expression of a series of transcription factors involved in osteocalcin regulation was performed in DPSCs ± knockdown for HDAC2 – however that data is not shown. This should be included in the manuscript as it was used as key argument to investigate GC.
9. In Figure 4B a more overview picture should be shown in addition – argumentation is done on only 4-5 nuclei shown. It would also increase the value of the manuscript to quantify (co-localization assay) the staining.
10. In Figure 5 the house-keepers seem to be regulated/uneven. Especially in the nuclear fraction the stated differences seem to be mainly due to the differences in the house-keeper/loading-control.
11. The authors might consider showing the results from the GPMiner analysis – as this is stated as method but not shown in the results.
12. Annotations in Figure 7 are unfortunately not readable. For FACS analysis results of the performed runs should be summarized (AUC or positive cells).
13. Although only supplementary figure - M&M for the cell cycle analysis should be given.
14. English writing seems to be done by several authors – Introduction and discussion are easy to follow – But results section is sometimes very complicated to read. This should be adjusted to allow fluent reading. Furthermore, authors should stick to either British or American English.
15. More references using VPA in osteogenic differentiation should be included in the manuscript (Discussion).
Author Response
February 6th, 2019
Dear Editor-in-Chief,
enclosed please find the revised form of the manuscript entitled “Cytoplasmic Interaction between Glucocorticoid Receptor and HDAC2 regulates Osteocalcin expression in VPA-treated MSCs”.
We have answered to each point raised by the referees #1, #2 and #3. The detailed answers are attached below this letter in red and, also in the manuscript, all corrections are in red.
Therefore, having answered to all the points, we hope that the manuscript, in its present revised form, will be now suitable for publication in “Cells”.
Yours faithfully,
With best regards,
Gianpaolo Papaccio Prof. MD., PhD
Reviewer #3:
We thank the reviewer# 3 for the constructive and helpful suggestions.
We have provided our point-by-point responses to the reviewer’s comments:
1.The link to HDAC2 is not clearly shown, as the used inhibitors should all inhibit HDAC2 activity.
R. In our previous paper (Paino et al., Stem cell 2014), we demonstrated that HDAC2 silencing led to an increased expression of OPN and BSP, but downregulated OC mRNA, resembling the effect of VPA. However, this downregulation did not occur when we analyzed the co-silencing of HDAC1 and HDAC2 or the silencing of HDAC1 alone. Therefore, the inhibition of OC gene expression by VPA is most likely mediated through HDAC2 inhibition. Starting from this assumption, we have compared the activity of VPA to the silencing of HDAC2. The other inhibitors used, probably, do not act specifically on HDAC2, being non-selective only for this enzyme. The VPA effect could be due to the fact that, in addition to selectively inhibiting the catalytic activity of class I HDACs, VPA induces proteasomal degradation of HDAC2 (Krämer OH et al, EMBO J.2003).
HDACi are classified on the basis of their chemical structure, and inhibit the enzymatic activity of the HDAC with different efficiencies and specificities. Hydroxamic acids, such as TSA and SAHA, are pan-inhibitors; the short-chain fatty acid, Valproic acid (VPA), in contrast, selectively inhibits HDAC of class I (HDAC1, 2,3,8); the benzamide MS-275, in turn, is selective only to a subclass of HDAC class I (preferentially inhibits HDAC1 and 2 at low doses as we considered, and has no inhibitory effect against HDAC8) (Hu E et al, J Pharmacol Exp Ther.2003; Khan N et al, Biochem J. 2008).Thus, evaluating the effect of each inhibitor on osteogenic differentiation we have determined the specific roles of individual HDACs.
However, we added major information about inhibitors used in Introduction and in Discussion as follows:
Line 66: The critical issue for application of stem cells in tissue engineering is the control of cell differentiation. The regulation of mechanisms underlining the stem cell state and/or differentiation is essential for the development of stem cell-based therapy. The events that govern stem cell differentiation are predominantly epigenetics. Indeed, specific patterns of DNA methylation and histone modifications play an important role in the induction of MSC differentiation toward specific lineages. Histone acetylation is the most well-studied histone modification and has been shown to be an important process of gene regulation. Histone acetylation contributes to the formation of a more relaxed and thus transcriptionally active chromatin structure. In contrast, histone deacetylation is associated with a condensed chromatin leading to transcriptional repression. The level of cellular histone acetylation is regulated by the opposing activities of histone acetyltransferases (HAT) and histone deacetylases (HDACs), which reversibly catalyze the acetylation and deacetylation mechanisms, respectively [20, 21]. HDACs have been shown to regulate several important biological processes, including cell proliferation, differentiation, and development, by forming complexes with various transcription factors and transcriptional coregulators [22]. The inhibition of HDAC activity typically leads to de-repression of transcription. HDAC inhibitors (HDACis) are natural or synthetic small molecules that can inhibit the activities of HDACs and promote efficient and temporally regulated control of gene expression. HDACis have shown very potent effects in stem cell differentiation pathways. They may promote either self-renewal [23;24] or differentiation [25;26] depending on the stem cell status, the dose employed and the cell type [27]. Therefore, application of epigenetic regulators, such as HDAC inhibitors, may be valuable for stem cell‐based interventions. There is growing evidence that some HDACi also stimulate osteogenic differentiation of MSCs [28-32].
Line 425: HDACi are classified on the basis of their chemical structure, and inhibit the enzymatic activity of the HDAC with different efficiencies and specificities. Hydroxamic acids, such as TSA and SAHA, are pan-inhibitors; the short-chain fatty acid, Valproic acid (VPA), in contrast, selectively inhibits HDAC of class I (HDAC1, 2,3,8); the benzamide MS-275, in turn, is selective only to a subclass of HDAC class I (preferentially inhibits HDAC1 and 2 at low doses as we considered [53,54]. Among them, Valproic acid (VPA)—an FDA-approved short-chain fatty acid—has been widely used for more than 20 years for the treatment of different neurological disorders [55]. VPA acts as a potent HDAC inhibitor at the concentrations used clinically [56]. Therefore, since the safety of VPA in patients has been already tested, its clinical use for new molecular approaches is of great interest.
In addition, treatment with the other inhibitors does not result in a reduction of the osteocalcin expression, which occurs only following treatment with VPA.
20. Legube, G., Legube, D. Regulating histone acetyltransferases and deacetylases. EMBO Rep.; 2003, 4:944–947.
21. Kuo, M.H., Allis, C.D. Roles of histone acetyltransferases and deacetylases in gene regulation. Bioessays.; 1998, 20:615–626
22. Haberland, M., Montgomery, R.L., Olson, E.N. The many roles of histone deacetylases in development and physiology: Implications for disease and therapy. Nature Reviews Genetics, 2009, 10:32–42
23. Ware, C.B., Wang, L., Mecham, B.H., et al. Histone deacetylase inhibition elicits an evolutionarily conserved self-renewal program in embryonic stem cells. Cell Stem Cell. 2009;4:359–369
24. Hayashi, K., Lopes, S.M.C.D.S., Tang, F., Surani, M.A. Dynamic equilibrium and heterogeneity of mouse pluripotent stem cells with distinct functional and epigenetic states. Cell Stem Cell. 2008;3:391–401
25. McCool, K.W., Xu, X., Singer, D.B., Murdoch, F.E., Fritsch, M.K. The role of histone acetylation in regulating early gene expression patterns during early embryonic stem cell differentiation. Journal of Biological Chemistry. 2007;282:6696–6706.
26. Karantzali, E., Schulz, H., Hummel, O., Hubner, N., Hatzopoulos, A.K., Kretsovali, A. Histone deacetylase inhibition accelerates the early events of stem cell differentiation: transcriptomic and epigenetic analysis. Genome Biology. 2008
27. Kretsovali, A., Hadjimichael, C., Charmpilas, N. Histone deacetylase inhibitors in cell pluripotency, differentiation, and reprogramming. Stem Cells Int. 2012;2012:184154.
28. Lee, S., Park, J.R., Seo, M.S., Roh, K.H., Park, S.B., Hwang, J.W., et al. Histone deacetylase inhibitors decrease proliferation potential and multilineage differentiation capability of human mesenchymal stem cells. Cell Prolif. 2009;42:711–20
29. Schroeder, T.M., Westendorf, J,J. Histone deacetylase inhibitors promote osteoblast maturation. J Bone Miner Res. 2005;20:2254–63
30. de Boer, J., Licht, R., Bongers, M., van der Klundert, T., Arends, R., van Blitterswijk, C. Inhibition of histone acetylation as a tool in bone tissue engineering. Tissue Eng. 2006;12:2927–37
31. Cho HH, Park HT, Kim YJ, Bae YC, Suh KT, Jung JS. Induction of osteogenic differentiation of human mesenchymal stem cells by histone deacetylase inhibitors. J Cell Biochem. 2005;96:533–42
32. Lee, H.W., Suh, J.H., Kim, A.Y., Lee, Y.S., Park, S.Y., Kim, J.B. Histone deacetylase 1-mediated histone modification regulates osteoblast differentiation. Mol Endocrinol. 2006;20:2432–43
48. Hu, E., Dul, E., Sung, C.M., Chen, Z., Kirkpatrick, R., Zhang, G.F., Johanson, K., Liu, R., Lago, A., Hofmann, G., Macarron, R., de los Frailes, M., Perez, P., Krawiec, J., Winkler, J., Jaye, M. Identification of novel isoform-selective inhibitors within class I histone deacetylases. J Pharmacol Exp Ther. 2003; 307:720-8.
49. Khan, N., Jeffers, M., Kumar, S., Hackett, C., Boldog, F., Khramtsov, N., Qian, X., Mills, E., Berghs, S.C., Carey, N., Finn, P.W., Collins, L.S., Tumber, A., Ritchie, J.W., Jensen, P.B., Lichenstein, H.S., Sehested, M. Determination of the class and isoform selectivity of small-molecule histone deacetylase inhibitors. Biochem J. 2008;409(2):581-9
50. Loscher, W. Basic pharmacology of valproate: A review after 35 years of clinical use for the treatment of epilepsy. CNS Drugs. 2002;16:669–694
51. Göttlicher, M., Minucci, S., Zhu, P., et al. Valproic acid defines a novel class of HDAC inhibitors inducing differentiation of transformed cells. EMBO J. 2001;20:6969–6978.
53. Ocker, M. Deacetylase inhibitors—focus on non-histone targets and effects. The World Journal of Biological Chemistry; 2010, 1(5):55–61.
54. Miyoshi, N., Ishii, H., Nagano, H., Haraguchi, N., Dewi, D.L., Kano, Y., Nishikawa, S., Tanemura, M., Mimori, K., Tanaka, F., et al. Reprogramming of mouse and human cells to pluripotency using mature microRNAs. Cell Stem Cell; 2011, 8(6):633–638.
55. Watari, S., Hayashi, K., Wood, J.A., Russell, P., Nealey, P.F., Murphy, C.J., Genetos, D.C. Modulation of osteogenic differentiation in hMSCs cells by submicron topographically-patterned ridges and grooves. Biomaterials; 2010, 31:3244-3252.
56. Naddeo, P., Laino, L., La Noce, M., Piattelli, A., De Rosa, A., Iezzi, G., Laino, G., Paino, F., Papaccio, G., Tirino, V. Surface biocompatibility of differently textured titanium implants with mesenchymal stem cells. Dent Mater; 2015, 31(3):235-43.
2. Furthermore, OC expression was investigated as VPA treated cell-constructs showed very good bone structure after subcutaneous culture – despite low OC levels. Final proof that increasing OC levels (stated with RU-486) in these cell-constructs gives comparable bone structure is missing.
R. We agree with reviewer about this missing point, that will be object of a new study. In this case, we used RU-486 only as an antagonist of GR, to confirm the involvement of this receptor in the reduction of osteocalcin expression after VPA treatment. Moreover, we postulate that is the osteocalcin downregulation responsible for more bone formation as it has been published by Ducy Pet al, Nature 1993. This hypothesis has to be confirmed by further investigations.
3. Using qRT-PCR gene identification (NM_ID), amplicon size, melting curves (selectivity of the method), and efficiency (standard curve) should be given. Furthermore, it should be shown that the used house-keeper was evenly expressed – as the observed differences in gene expression are rather low.
R. We have added the requested information in Table 1.
GENE | Accession number | Sequence | Amplicon size (bp) | Tm (°C) | Ta (°C) |
GAPDH | NM_002046.7 | FW: ggagtcaacggatttggtcg REV: cttcccgttctcagccttga | 180 | 81.7 | 60 |
OC | NM_199173.6 | FW: ctcacactcctcgccctattg REV: cttggacacaaaggctgcac | 108 | 87.51 | 60 |
OPN | NM_001040058.2 | FW: gccgaggtgatagtgtggtt REV: tgaggtgatgtcctcgtctg | 101 | 79.33 | 58 |
BSP | NM_004967.4 | FW: ctggcacagggtatacagggttag REV: actggtgccgtttatgccttg | 182 | 79.46 | 60 |
GR | NM_000176.3 | FW: ggagaggggagatgtgatgg REV: agggtgaagacgcagaaacc | 72 | 79.16 | 60 |
4. In Materials and Methods IF staining against anti-human nuclear antigen is noted but not shown in the results section. Also it should be named from which species the AB (mono- or polyclonal) are. Furthermore, background controls (only secondary AB should be shown)
R. We apologize for not having specified this in the results. We agree with the reviewer and added the IF image of Human Nuclear Antigen in the supplemental data, as we predicted, to demonstrate the presence of human cells in the explants (Fig. 3S).
Please, see figure in the attached file
We have added information about the antibody in material and methods section.
5.It is stated that Student’s T-test was used for statistics – this is only suitable for comparing two groups that are normally distributed. Here more groups are compared thus an ANOVA based test has to be used. Based on the small sample size this test should be non-parametric.
R. We apologize for this mistake. We have reconsidered the statistical analysis and correct the paragraph in material and methods section:
Line 271: All experiments were carried out in triplicate and repeated at least three times. The results were presented as means ± SEM with p ≤ 0.05 considered as statistically significant. Western blot images were semi-quantitatively analyzed with ImageJ software (NIH, USA). Statistical analysis was performed by analysis of variance (ANOVA) for multiple comparison test.
6.In the graphics it is not clear what statistics are shown: meaning of * should be given in the figure legends – but even more important: it should be clarified which data sets the stated * refers to.
R. We added in figure legends the meaning of *. *p < 0.05
7.Regulation of OC in figure 1 cannot be really seen – please consider changing the figure for clarification.
R. As suggested by the reviewer we have changed the figure to clarify the differences in gene expression in different treatments.
Please, see figure in the attached file
8. It was stated that expression of a series of transcription factors involved in osteocalcin regulation was performed in DPSCs ± knockdown for HDAC2 – however that data is not shown. This should be included in the manuscript as it was used as key argument to investigate GC
R. We evaluated many transcription factors, either by using an array, or by using qRT-PCR analysis, but none of them showed a variation in gene expression and correlation with VPA treatment. Studies on osteocalcin expression were not associated with any of them, except for GR. Therefore, we have chosen not to show the data as no to make the manuscript hard to read and understand.
9.In Figure 4B a more overview picture should be shown in addition – argumentation is done on only 4-5 nuclei shown. It would also increase the value of the manuscript to quantify (co-localization assay) the staining.
R. We agree with reviewer’s suggestion, but to better show the differences of expression and the cellular localization we used a magnification of 100X that does not allow to visualize a high number of cells. However, we send the reviewer a different image to confirm what was shown in Figure 4B.
Please, see figure in the attached file
10. In Figure 5 the house-keepers seem to be regulated/uneven. Especially in the nuclear fraction the stated differences seem to be mainly due to the differences in the house-keeper/loading-control.
R. We changed the housekeeper bands, and the quantification by image J.
Please, see figure in the attached file
11. The authors might consider showing the results from the GPMiner analysis – as this is stated as method but not shown in the results.
R. GPMiner was used as a software in which we inserted the FASTA sequence of the osteocalcin promoter. It showed us the sites of Glucocorticoid Receptor Element in the promoter of the osteocalcin, from which we disegned the specific primers. We added the link of the analisys in results section as follows:
Line 386: The results showed that GR predominantly binds on nGRE sequence that we identified by GPMiner (http://gpminer.mbc.nctu.edu.tw/show_prediction/show.php?OS=human&ID=20180110_193635&scale=3&GC_window=15&TFBS_core=1.00&TFBS_matrix=0.95&OR_zscore=5&OR_number=20&OR_occur=2&stability_window=15&miRNA_MFE=&miRNA_score=)...
12.Annotations in Figure 7 are unfortunately not readable. For FACS analysis results of the performed runs should be summarized (AUC or positive cells).
R. We have enlarged the annotations to make them more readable. Moreover, we added histograms in figure 7 to summarize the results.
Please, see figure in the attached file
13.Although only supplementary figure - M&M for the cell cycle analysis should be given
R. We apologize for this mistake, we added M&M for cell cycle:
Line 266: For cell cycle analysis, cells were detached by trypsinization and then fixed with ice-cold 80% ethanol. The cells were centrifuged and then stained with a solution of 50 µg/mL propidium iodide and 80 µg/mL RNase A for 60 min at 4 °C in the dark. DNA content and cell cycle distribution were measured with a FACSARIA III. All data analyzed with FCS Express version 3 software.
14.English writing seems to be done by several authors – Introduction and discussion are easy to follow – But results section is sometimes very complicated to read. This should be adjusted to allow fluent reading. Furthermore, authors should stick to either British or American English.
R. We reviewed and adjusted results section
15. More references using VPA in osteogenic differentiation should be included in the manuscript (Discussion).
R. We have included more references in the Discussion section as previously answered (R1).
Line 425: HDACi are classified on the basis of their chemical structure, and inhibit the enzymatic activity of the HDAC with different efficiencies and specificities. Hydroxamic acids, such as TSA and SAHA, are pan-inhibitors; the short-chain fatty acid, Valproic acid (VPA), in contrast, selectively inhibits HDAC of class I (HDAC1, 2,3,8); the benzamide MS-275, in turn, is selective only to a subclass of HDAC class I (preferentially inhibits HDAC1 and 2 at low doses as we considered [53,54]. Among them, Valproic acid (VPA)—an FDA-approved short-chain fatty acid—has been widely used for more than 20 years for the treatment of different neurological disorders [55]. VPA acts as a potent HDAC inhibitor at the concentrations used clinically [56]. Therefore, since the safety of VPA in patients has been already tested, its clinical use for new molecular approaches is of great interest.
53. Ocker, M. Deacetylase inhibitors—focus on non-histone targets and effects. The World Journal of Biological Chemistry; 2010, 1(5):55–61.
54. Miyoshi, N., Ishii, H., Nagano, H., Haraguchi, N., Dewi, D.L., Kano, Y., Nishikawa, S., Tanemura, M., Mimori, K., Tanaka, F., et al. Reprogramming of mouse and human cells to pluripotency using mature microRNAs. Cell Stem Cell; 2011, 8(6):633–638.
55. Watari, S., Hayashi, K., Wood, J.A., Russell, P., Nealey, P.F., Murphy, C.J., Genetos, D.C. Modulation of osteogenic differentiation in hMSCs cells by submicron topographically-patterned ridges and grooves. Biomaterials; 2010, 31:3244-3252.
56. Naddeo, P., Laino, L., La Noce, M., Piattelli, A., De Rosa, A., Iezzi, G., Laino, G., Paino, F., Papaccio, G., Tirino, V. Surface biocompatibility of differently textured titanium implants with mesenchymal stem cells. Dent Mater; 2015, 31(3):235-43.

Round 2
Reviewer 1 Report
The authors answered appropriately my concerns and questions
Author Response
We thank this reviewer for his suggestion. In this form, the manuscript is improved.
Reviewer 3 Report
The revised manuscript “Cytoplasmic Interaction between Glucocorticoid Receptor and HDAC2 regulates Osteocalcin expression in VPA-treated MSCs” from Marcella La Noce et al. has improved. However, some questions remained un-answered.
1. I really appreciate the work the authros have put in adding information on the HDAC inhibitors used – this helps a lot to follow the manuscript. The link to HDAC2 exclusively remains unclear – but is much better discussed now. The most interesting part is the proposed proteasomal degradation of HDAC2 VPA should induce – this could have been shown but might have extended the limits of this study.
2. “Final proof that increasing OC levels (stated with RU-486) in these cell-constructs gives comparable bone structure is missing” - If this point cannot be shown, the conclusions drawn should be softened / drawn more carefully.
3. Using qRT-PCR gene identification melting curves (selectivity of the method) and efficiency (standard curve) should be given. Furthermore, it should be shown that the used house-keeper was evenly expressed – this information is still missing.
4. Background controls (only secondary AB) of the fluorescent images should be shown – this information is still missing
5. Thank you for stating that another statistical test was used – please clarify which form of ANOVA was used and with which post-hoc test the statistics were obtained.
6. Thank you for adding * < 0.05 in the figure legends – please also state to which reference group this belongs
7. Figure 6: Please clarify which data sets the stated * refers to.
8. Regulation of OC in figure 1 still cannot be really seen – please consider changing the figure to A, B, C with individual y-axis for clarification.
9. It was stated that expression of a series of transcription factors involved in osteocalcin regulation was performed in DPSCs ± knockdown for HDAC2 – however that data is not shown. Please include the data in the manuscript (supplementary) as this was used as key argument to investigate GC.
10. Thank you for providing new house-keeping pictures for Figure 5 – please double check the densitometric analysis for figure 5B as the image of the bands does not seem to fit to the bar chart.
11. Thank you for including the link to the GPMiner analysis – unfortunately the link seems corrupt – please double check.
12. Introduction and discussion are easy to follow – But results section is sometimes very complicated to read. – The authors mentioned that this was adjusted but unfortunately this is not comprehensible.
13. Thank you for providing more information on the actual HDAC inhibitors used – but as mentioned before more references using VPA in osteogenic differentiation should be included in the manuscript (Discussion), e.g.:
a. Triantafyllou N, et al.: Effect of long-term valproate monotherapy on bone mineral density in adults with epilepsy. J Neurol Sci 2010
b. Ehnert S, et al: Transforming growth factor beta1 inhibits bone morphogenic protein (BMP)-2 and BMP-7 signaling via upregulation of Ski-related novel protein N (SnoN): possible mechanism for the failure of BMP therapy? BMC medicine 2012
c. Maroni P, et al.: Chemical and genetic blockade of HDACs enhances osteogenic differentiation of human adipose tissue-derived stem cells by oppositely affecting osteogenic and adipogenic transcription factors. Biochem Biophys Res Commun 2012
d. Schroeder TM, Westendorf JJ: Histone deacetylase inhibitors promote osteoblast maturation. J Bone Miner Res 2005
Author Response
1. I really appreciate the work the authors have put in adding information on the HDAC inhibitors used – this helps a lot to follow the manuscript. The link to HDAC2 exclusively remains unclear – but is much better discussed now. The most interesting part is the proposed proteasomal degradation of HDAC2 VPA should induce – this could have been shown but might have extended the limits of this study.
We thank this reviewer for appreciation. Actually, we have shown the link between VPA and HDAC2 in our previous work (Paino et al, Stem Cell). Therefore, this study was focused on the molecular mechanism involved in reducing osteocalcin expression. We have demonstrated that the GR translocates in the nucleus and inhibits the expression of the osteocalcin after VPA treatment.
2. “Final proof that increasing OC levels (stated with RU-486) in these cell-constructs gives comparable bone structure is missing” - If this point cannot be shown, the conclusions drawn should be softened / drawn more carefully.
We thank reviewer for this suggestion. As previously stated, the main aim of this study was to investigate the molecular mechanisms undergoing osteocalcin reduction. To this end, we used RU-486 only as an antagonist of GR, to confirm the involvement of this receptor in the reduction of osteocalcin expression after VPA treatment. Therefore, we have better explained this in Result section and softened the conclusions, as suggested.
3. Using qRT-PCR gene identification melting curves (selectivity of the method) and efficiency(standard curve) should be given. Furthermore, it should be shown that the used house-keeper was evenly expressed – this information is still missing.
We added information about efficiency and melting curves in M&M section. Moreover, we send to this reviewer images to show melting curves and that the house-keeping gene was evenly expressed, as shown by the similarity of Ct.
4. Background controls (only secondary AB) of the fluorescent images should be shown – this information is still missing.
We apologize for this mistake. We added background controls images in supplemental figure 4.
5. Thank you for stating that another statistical test was used – please clarify which form of ANOVA was used and with which post-hoc test the statistics were obtained.
We apologize for this mistake. We added the information required in M&M section.
6. Thank you for adding * < 0.05 in the figure legends – please also state to which reference group this belongs.
We apologize for this mistake. We added the reference group to which it refers in the figure legend.
7. Figure 6: Please clarify which data sets the stated * refers to.
We modified figure 6 and added to which group * refers in the legend.
8. Regulation of OC in figure 1 still cannot be really seen – please consider changing the figure to A, B, C with individual y-axis for clarification.
We modified figure 1 as suggested.
9. It was stated that expression of a series of transcription factors involved in osteocalcin regulation was performed in DPSCs ± knockdown for HDAC2 – however that data is not shown. Please include the data in the manuscript (supplementary) as this was used as key argument to investigate GC.
As we stated in our previous revision, we evaluated many transcription factors and genes and we added in Material and Methods the array and genes we analyzed for this purpose. Moreover, we amended the Result Section to make it more understandable. In addition we show in Suppl. fig. 5 the results of each gene expression.
10. Thank you for providing new house-keeping pictures for Figure 5 – please double check the densitometric analysis for figure 5B as the image of the bands does not seem to fit to the bar chart.
We re-performed densitometric analysis for figure 5B and changed the histogram.
11. Thank you for including the link to the GPMiner analysis – unfortunately the link seems corrupt – please double check.
We adjusted the link to GPMiner.
12. Introduction and discussion are easy to follow – But results section is sometimes very complicated to read. – The authors mentioned that this was adjusted but unfortunately this is not comprehensible.
We apologize for this mistake. We modified results section.
13. Thank you for providing more information on the actual HDAC inhibitors used – but as mentioned before more references using VPA in osteogenic differentiation should be included in the manuscript (Discussion), e.g.:
a. Triantafyllou N, et al.: Effect of long-term valproate monotherapy on bone mineral density in adults with epilepsy. J Neurol Sci 2010
b. Ehnert S, et al: Transforming growth factor beta1 inhibits bone morphogenic protein (BMP)-2 and BMP-7 signaling via upregulation of Ski-related novel protein N (SnoN): possible mechanism for the failure of BMP therapy? BMC medicine 2012
c. Maroni P, et al.: Chemical and genetic blockade of HDACs enhances osteogenic differentiation of human adipose tissue-derived stem cells by oppositely affecting osteogenic and adipogenic transcription factors. Biochem Biophys Res Commun 2012
d. Schroeder TM, Westendorf JJ: Histone deacetylase inhibitors promote osteoblast maturation. J Bone Miner Res 2005
We added the references about the use of VPA in osteogenic differentiation in the Discussion section, as suggested by this reviewer. The reference “Schroeder et al.” was already present in the paper (29).
